# Replicable Online Learning

Saba Ahmadi*

Toyota Technological Institute at Chicago
`saba@ttic.edu`
& Siddharth Bhandari
Toyota Technological Institute at Chicago
`siddharth@ttic.edu`
& Avrim Blum
Toyota Technological Institute at Chicago
`avrim@ttic.edu`

## Abstract

We investigate the concept of algorithmic replicability introduced by Impagliazzo et al. [2022], Ghazi et al. [2021], Ahn et al. [2024] in an online setting. In our model, the input sequence received by the online learner is generated from *time-varying distributions* chosen by an adversary (obliviously). Our objective is to design low-regret online algorithms that, with high probability, produce the *exact same sequence* of actions when run on two independently sampled input sequences generated as described above. We refer to such algorithms as adversarially replicable.

Previous works (such as Esfandiari et al. [2022]) explored replicability in the online setting under inputs generated independently from a fixed distribution; we term this notion as iid-replicability. Our model generalizes to capture both adversarial and iid input sequences, as well as their mixtures, which can be modeled by setting certain distributions as point-masses.

We demonstrate adversarially replicable online learning algorithms for online linear optimization and the experts problem that achieve sub-linear regret. Additionally, we propose a general framework for converting an online learner into an adversarially replicable one within our setting, bounding the new regret in terms of the original algorithms regret. We also present a nearly optimal (in terms of regret) iid-replicable online algorithm for the experts problem, highlighting the distinction between the iid and adversarial notions of replicability.

Finally, we establish lower bounds on the regret (in terms of the replicability parameter and time) that any replicable online algorithm must incur.

## 1 Introduction

The replicability crisis, which is pervasive across scientific disciplines, has substantial implications for the integrity and reliability of findings. In a survey of 1,500 researchers, it was reported that 70% had attempted but failed to replicate another researchers findings [Baker, 2016]. A recent Nature article [Ball, 2023] discusses how the replicability crisis in AI is creating a ripple effect across numerous scientific fields, including medicine, due to AIs broad applications. This highlights an urgent need to establish a formal framework for assessing the replicability of experiments in machine learning.

Driven by this context, Impagliazzo et al. [2022] (see also Ghazi et al. [2021], Ahn et al. [2024]) initiate the study of reproducibility as a property of algorithms themselves, rather than the process

---

*Authors are ordered alphabetically. The second author did this work prior to joining Amazon.

39th Conference on Neural Information Processing Systems (NeurIPS 2025).

by which their results are collected and reported. An algorithm ALG drawing samples from an (unknown) distribution $\mathcal{D}$ is called $\rho$-replicable if, run twice on independent samples and the *same randomness R*, ALG produces *exactly the same answer* with probability at least $1 - \rho$. For instance, consider the task of estimating the average value of a distribution $\mathcal{D}$ over $\mathbb{R}$, say $\mu$, using $n$ iid sample from $\mathcal{D}$. If we simply use the empirical average, say $\hat{\mu}$, it won't lead to a $\rho$-replicable algorithm. However, replicability can be achieved at the cost of a little accuracy: we overlay a grid on $\mathbb{R}$ with a fixed width (sufficiently larger than the standard deviation of $\hat{\mu}$) but a random offset, and then round $\hat{\mu}$ to the nearest grid point (see Impagliazzo et al. [2022] for more details). In this work, we focus on studying and designing replicable online learning algorithms.

In the context of an online learning algorithm, say for instance for the setting of learning with experts advice or online linear optimization, $\rho$-replicability means that when the algorithm is run using the same internal randomness on two input sequences sampled iid from an (unknown) distribution $\mathcal{D}$, the *entire sequence of actions* performed is identical in both the runs with probability at least $1 - \rho$. More precisely, suppose the inputs to the an online algorithm, say ALG, arrive one by one, i.e., input $c_t$ at time $t$, where $c_t \in \chi$ for some input domain $\chi$. For an input sequence $S = (c_1, c_2, \ldots, c_T)$ we let $a_t(S, R)$ be the action of ALG at time $t$: since ALG is an online algorithm, $a_t(S, R)$ is a function only of the history seen till time $t - 1$ and internal randomness $R$. We say that ALG is $\rho$-replicable if for all input distributions $\mathcal{D}$ over $\chi$ we have:

$$\Pr_{\substack{S \leftarrow \mathcal{D}^{\otimes T} \\ S' \leftarrow \mathcal{D}^{\otimes T} \\ R}} [\forall t : a_t(S, R) = a_t(S', R)] \geq 1 - \rho,$$

where $S$ and $S'$ are independent sequences of $T$ iid random variables sampled from $\mathcal{D}$ (see Section 2).

However, we would also like our online algorithm to achieve low regret as compared to the best course of action in hindsight. In the above set-up let the cost incurred by an action $a \in \mathcal{A}$, where $\mathcal{A}$ is the action space, on input $c$ be $\mathsf{Cost}(a, c)$. For instance, in the case of online linear optimization $a, c \in \mathbb{R}^n$ and $\mathsf{Cost}(a, c) = a \cdot c$ (the inner product of $a$ and $c$), and for the experts setting $a \in [n], c \in \mathbb{R}^n$ and $\mathsf{Cost}(a, c) = c(a)$ the cost of expert $a$. Then, the regret of ALG on input $S = (c_1, \ldots, c_T)$ is

$$\mathsf{Reg}(\mathsf{ALG}(S)) := \mathbb{E}_R \left[ \sum_{t=1}^{T} \mathsf{Cost}(a_t, c_t) \right] - \min_{a \in \mathcal{A}} \sum_{t=1}^{T} \mathsf{Cost}(a, c_t).$$

Since the notion of replicability is under stochastic iid inputs, a first demand on ALG is that while being $\rho$-replicable, it has low expected regret when the input sequence is sampled iid from $\mathcal{D}$, for all input distributions $\mathcal{D}$. In other words, $\mathbb{E}_{S \leftarrow \mathcal{D}^{\otimes T}}[\mathsf{Reg}(\mathsf{ALG}(S))]$ is low and ALG is $\rho$-replicable. Such a setting was explored by Esfandiari et al. [2022] in the case of stochastic multi-arm bandits: see Section A for more works in this setting . A stronger yet natural requirement is that the worst-case regret of ALG remains low; that is, $\sup_{S \in \chi^T} \mathsf{Reg}(\mathsf{ALG}(S))$ is minimized while maintaining $\rho$-replicability. This gives us a best-of-both-worlds scenario: the algorithm achieves low regret in the worst case, while remaining replicable for stochastic iid inputs.

Another natural extension is to allow the distribution on inputs to vary over time. Since an online algorithm processes inputs indexed by time, it is reasonable to permit their distributions to be time-dependent. For instance, consider an online algorithm that recommends a stock to invest in daily based on historical performance, or the online shortest path problem in traffic management, where edge weights at time $t$ represent travel times on streets. One could also generalize this requirement to cases where part of the input is stochastic and part is adversarial.

We capture the above points into a single concept. We assume that for each time $t$ the adversary chooses a distribution $\mathcal{D}_t$ (in an oblivious fashion) and selects a cost vector $c_t$ from that distribution. The distributions $\mathcal{D}_t$ could be point-masses, reducing to the standard adversarial online formulation, but they could also be more general. The goal of the online learner ALG is to (a) achieve low regret with respect to the actual sequence $c_1, c_2, ..., c_T$ of cost vectors observed and (b) to be replicable: for any sequence $\mathcal{D}_1, \mathcal{D}_2, ... \mathcal{D}_T$, with probability at least $1 - \rho$ the algorithm has the exact same behavior on two different draws $S, S'$ from this sequence when the learning algorithm uses the same randomness $R$, i.e.,

$$\Pr_{S, S', R} [\forall t : a_t(S, R) = a_t(S', R)] \geq 1 - \rho.$$

If this is the case then we say that ALG is adversarially $\rho$-replicable. To distinguish it from the previous situation we will say that ALG is iid $\rho$-replicable if the replicability guarantees are only against iid inputs. (See Section 2.)

**Remark.** *One could also study a notion of approximate replicability by requiring that, for each time step $t$,*

$$\Pr_{S,S',R} [a_t(S, R) = a_t(S', R)] \geq 1 - \rho,$$

*meaning that at each time step $t$, the actions of the algorithm on $S$ and $S'$ are the same with probability at least $1 - \rho$. This contrasts with the stronger requirement that the entire sequence of actions on $S$ and $S'$ be identical with probability at least $1 - \rho$.*

*Choosing the stronger notion of replicability of course implies approximate replicability, but it also allows the algorithm to be composed with other algorithms, preserving replicability across compositions.*

## 1.1 Our Contributions

We present online learning algorithms for online linear optimization and the experts problem that achieve both sub-linear regret and adversarial $\rho$-replicability. Moreover, we introduce a general framework for transforming an online learner for the above problems into an adversarial $\rho$-replicable online learner, with a bound on its regret based on the regret of the original algorithm. Furthermore, we also give an almost optimal algorithm (in terms of regret) for the experts problem in the iid-replicability setting.

We also explore how replicability affects the regret of an algorithm by proving lower bounds of the regret of any online $\rho$-replicable algorithm in the adversarial and iid settings. Up to lower order terms the upper and lower bounds match for the iid-replicability setting while their being a gap for the adversarial-replicability setting.

The key ideas are outlined below.

### 1.1.1 Adversarially $\rho$-replicable Online Linear Optimization

We examine the online linear optimization problem which is a linear generalization of the standard online learning problem. Here, at time $t$ the algorithm taken an action $a_t \in \mathbb{R}^n$ and receive an input $c_t \in \mathbb{R}^n$, and accrues a cost $a_t \cdot c_t$ (see Section 2). In this context, we demonstrate a low regret algorithm that is adversarially $\rho$-replicable. Our algorithm leverages two main strategies: a) partitioning the time horizon into blocks and updating its action only at the endpoints of blocks (referred to as transition points), and b) rounding of cumulative cost vectors to the grid points in a random grid. Below, we provide a brief overview of these key ideas:

**Blocking of time steps:** The algorithm partitions the time horizon into blocks of fixed size and only changes its actions at the end of a block. Here, for two independent sequences $S, S'$ drawn from the same sequence $\mathcal{D}_1, \cdots, \mathcal{D}_T$, since the sequences are close to each other but not exactly the same, a traditional low-regret algorithm, e.g. "Follow the Perturbed Leader"(FTPL) by Kalai and Vempala [2005], might change its actions in different but relevantly close time steps. Consequently, the idea of blocking allows us to argue that this switching behavior of the algorithm happens with high probability within the same block. Hence, when the algorithm only changes its actions at the endpoint of a block, with high probability it shows the same behavior over two different sequences.

**Rounding of cumulative cost vectors to the grid points in a random grid** Here, at the end of each block, the online learner rounds the cumulative cost vectors to the grid points in a grid with a random offset, and selects the action that minimizes the total cost pretending the grid point is the cost vector. This idea is inspired by the seminal "Follow the Lazy Leader" (FLL) algorithm of Kalai and Vempala [2005]. Since $S, S'$ are drawn from the same sequence $\mathcal{D}_1, \cdots, \mathcal{D}_T$, the cumulative cost vectors are within a bounded $\ell_1$ distance with high probability, and hence the cost vectors get rounded to the same grid point with high probability. This property helps us argue that the online learner picks the same action at all the block endpoints (and also all the time steps) with high probability.

These two ideas help us achieve replicability guarantees, however, we need to be careful when setting the block size and the randomness of the grid; as we increase them, it is easier to achieve

replicability, however, the regret will suffer. We show that by carefully selecting the blocksize and picking the randomness of the grid, low-regret learning and replicability over all time steps is possible.

**Theorem 1.1** (Informal; see Theorem 3.1). *Let $\rho > 0$ be a parameter. For the online linear optimization problem, there exists an adversarially $\rho$-replicable algorithm with sub-linear regret.*

### 1.1.2 An Adversarially $\rho$-Replicable Algorithm for the Experts Problem

Next, we explore the experts problem, where, in each period, the online learner selects an expert from a set of n experts, incurs the cost associated with the chosen expert, and observes the cost $[0, 1]$ for all experts in that round. The objective of the online learner is to achieve low regret compared to the best expert in hindsight while ensuring replicability across all time steps. We present an adversarially $\rho$-replicable learning algorithm with sub-linear regret. This algorithm applies the strategy of partitioning the time horizon into blocks and only updating its actions at the end of each block. Initially, geometric noise is added to each expert, and at the end of each block, the algorithm chooses the expert with the lowest cumulative cost, incorporating the initial noise. We demonstrate that, with appropriately chosen block sizes and noise levels, it is possible to achieve low-regret learning along with adversarial $\rho$-replicability. To prove replicability, we leverage the properties of geometric noise, such as memorylessness, and argue that when two trajectories are sufficiently close in $\ell_\infty$ distance, with high probability the best experts in these trajectories are the same.

**Theorem 1.2** (Informal; see Theorem 4.1). *Let $\rho > 0$ be a parameter. For the experts problem, there exists an adversarially $\rho$-replicable algorithm with sub-linear regret.*

### 1.1.3 A General Framework for Converting an Online Learning Algorithm to an Adversarially $\rho$-Replicable Learning Algorithm

Furthermore, in Section 5, we present a general framework for converting an online learner to an adversarially $\rho$-replicable algorithm where its regret is a function of the regret of the initial algorithm. For this conversion we need the further assumption that the cost incurred on action $a$ and input $c$, i.e., $\mathsf{Cost}(a, c)$, is linear in $c$: $\mathsf{Cost}(a, c_1 + c_2) = \mathsf{Cost}(a, c_1) + \mathsf{Cost}(a, c_2)$ (this assumes that the input sequence is itself in an ambient vector space). We illustrate how this general framework can be applied to both online linear optimization and expert problems. However, this approach comes with a trade-off: while it provides a versatile and unified solution, it gives worse regret bounds compared to algorithms specifically designed for each individual setting.

We continue with the approach of grouping time steps into blocks and rounding the cumulative cost vector at the end of each block to the nearest grid point. Specifically, at the end of block $i$, we compute the difference between the rounded cumulative cost vectors at the ends of blocks $i$ and $i - 1$. This difference is then fed as the $i$-th input to the initial algorithm, and the action suggested by the initial algorithm is applied throughout block $i + 1$.

However, we need a slightly more sophisticated approach to relate the regret bound of the external algorithm to that of the internal algorithm (the initial online learner). When providing the $i$-th input to the internal algorithm, we use two fresh random grids to compute the rounded cumulative cost vectors at the ends of blocks $i$ and $i - 1$. Consequently, the cumulative cost vector at the end of block $i$ is rounded in two different ways: once for the $i$-th input to the internal algorithm and once for the $(i + 1)$-th block. (See Lemma F.1 for more details on why this helps.)

As before, we need to strike a balance between the grid size and the block size to achieve both low-regret and adversarial replicability.

**Theorem 1.3** (Informal; see Theorem 5.1). *Let $\rho > 0$ be a parameter, and assume that the cost function $\mathsf{Cost}(a, c)$ is linear in $c$, i.e., $\mathsf{Cost}(a, c_1 + c_2) = \mathsf{Cost}(a, c_1) + \mathsf{Cost}(a, c_2)$. Given an internal algorithm $\mathsf{ALG}_{int}$ with bounded regret, it can be converted to an adversarially $\rho$-replicable external algorithm $\mathsf{ALG}_{ext}$ with bounded regret.*

### 1.1.4 iid-Replicability for the Experts Problem

To investigate and contrast the differences between iid and adversarial replicability, in Section H we design an iid $\rho$-replicable online algorithm for the experts problem which has low worst-case regret. Recall that in this setting, we have $n$ experts, and the cost vectors are drawn iid from an unknown distribution $\mathcal{D}$ over $[0, 1]^n$. The goal is to design an iid $\rho$-replicable algorithm with worst-case regret as low as possible which means the following: (1)We say an algorithm has worst-case regret $K$ if

for any cost sequence $S = (c_1, ..., c_T)$, the expected regret satisfies $\mathbb{E}_{R \sim \mathcal{R}}[Reg(S, R)]$ (2) We say an algorithm has replicability $\rho$ if for iid cost sequences $S$ and $S'$ we have

$$\Pr_{S, S', R}[\forall t : a_t(S, R) = a_t(S', R)] \geq 1 - \rho.$$

While it is possible to use algorithms that are adversarially replicable, we adopt a different approach here to achieve better regret bounds. We still group time steps into blocks, but in this case, the blocks are of varying lengths. Additionally, we add geometric noise of varying magnitude at the end of each block, which helps in keeping the regret low while achieving replicability.

As before, at the end of each block, we select the expert with the minimum cumulative cost (incorporating the added noise) to use for the entire next block. The first block has length approximately $\sqrt{T}$, providing an estimate of each experts expected cost under $\mathcal{D}$ with an accuracy of $T^{-1/4}$. Using these estimates, we select the best expert at the end of block 1 (after incorporating noise) and stick with them for the next block, which has length $T^{3/4}$. Proceeding in this manner, we ensure an expected regret of at most $\sqrt{T}$ per block, with at most $\log\log(T)$ blocks in total. (See Algorithm 9 for details.)

**Theorem 1.4** (Informal; see Theorem H.1). *Let $\rho > 0$ be a parameter. For the experts problem, there exists an iid $\rho$-replicable algorithm with worst-case regret roughly $O(\sqrt{T}\log(n)/\rho)$.*

In light of our lower bounds (see below or Theorem I.2) the regret achieved is optimal up to $poly(\log\log(T))$ and other logarithmic factors.

### 1.1.5   Lower Bounds on Regret for Replicable Online Algorithms

In Section I we prove lower bounds on the regret a $\rho$-replicable algorithm must suffer, both in the iid and adversarial settings. The main idea for the iid setting is to using the lower bound on the replicability of the coin problem from Impagliazzo et al. [2022]. The coin problem involves identifying the bias of a coin (either $1/2 + \tau$ or $1/2 - \tau$) using $T$ iid samples, with success probability $1 - \delta$ and replicability at least $1 - \rho$, which requires $\rho \geq \Omega\left(\frac{1}{\tau\sqrt{T}}\right)$. By embedding this problem in the experts setting (even with $n = 2$) with $\tau \approx \mathsf{Reg}/T$, we obtain $\mathsf{Reg} > \Omega\left(\frac{\sqrt{T}}{\rho}\right)$. In the case of adversarial replicability with $n$ experts, we use the above idea in $\log(n)$ phases of length $\approx T/\log(n)$ to essentially embed $\log(n)$ different instances of the coin problem.

**Theorem 1.5** (Informal; see Theorems I.2 and I.3). *If the parameter $\rho$ is sufficiently larger than $1/\sqrt{T}$ then any iid $\rho$-replicable algorithm must suffer a worst-case regret $\Omega(\sqrt{T}(1/\rho + \sqrt{\log(n)}))$. Further, any adversarial $\rho$-replicable algorithm must suffer a worst-case regret of $\Omega(\sqrt{T\log(n)}/\rho)$.*

## 2   Model

**Online Linear Optimization**   Online linear optimization is a linear extension of the online learning problem, where the online learner ALG must take a series of actions $a_1, a_2, \cdots$ each from a possibly infinite set of actions $\mathcal{A} \subset \mathbb{R}^n$. After the $t^{th}$ action is taken, the decision maker observes the current step's cost $c_t \in \mathcal{C} \subset \mathbb{R}^n$. The cost of taking an action $a$ when the cost is $c$ is $a \cdot c$, and the total cost incurred by the algorithm is $\sum_t a_t \cdot c_t$. We use $S = \{c_1, \cdots, c_T\}$ to denote the sequence of costs. The learner aims to minimize the regret accrued over $T$ steps where regret is defined as:

$$\mathsf{Reg}(\mathsf{ALG}(S)) = \sum_t c_t \cdot a_t - \min_{a \in \mathcal{A}} \sum_t a \cdot c_t$$

**Experts Problem**   In this problem, in each period $t$, the learner ALG picks an expert $a_t$ from a set of $n$ experts denoted by $\mathcal{A}$, pays the cost associated with the chosen expert $c_t(a_t)$, and then observes the cost between $[0, 1]$ for each expert. Let $S$ denote the cost sequence $\{c_1, \cdots, c_T\}$. The goal of the learner is to have low regret, i.e. ensure that its total cost is not much larger than the minimum total cost of any expert.

$$\mathsf{Reg}(\mathsf{ALG}(S)) = \sum_t c_t(a_t) - \min_{a \in \mathcal{A}} \sum_t c_t(a)$$

**iid-Replicability** Let $\mathcal{D}$ be a distribution over an input domain $\chi$, and let ALG be an online algorithm. Let $S, S'$ be independent sequences of length $T$ of inputs drawn iid from $\mathcal{D}$, and let $R$ represent the internal randomness used by ALG. Further, let $a_t(S, R)$ be the action at time $t$ on input $S$ and internal randomness $R$. We say that ALG is $\rho$-iid replicable if:

$$\Pr_{S,S',R}[\forall t \in [T] : a_t(S, R) = a_t(S', R)] \geq 1 - \rho.$$

**Adversarial Online Replicability** Throughout the paper, we consider a more general version of replicability where all the examples in the sequences $S, S'$ are not drawn necessarily from the same distribution $\mathcal{D}$. Instead, we assume an adversarial sequence of distributions $\mathcal{D}_1, \mathcal{D}_2, ..., \mathcal{D}_T$, and the $t^{th}$ element of the sequence is drawn from distribution $\mathcal{D}_t$. Let $\mathcal{P} = \otimes_{t=1}^{T} \mathcal{D}_t$, i.e., the product distribution $\mathcal{D}_1 \times \ldots \times \mathcal{D}_T$. Under this assumption, we say an online algorithm ALG is adversarially $\rho$-replicable if for sequences $S, S'$ each of length $T$ sampled independently from $\mathcal{P}$, it produces the same sequence of actions with probability at least $1 - \rho$, i.e.,

$$\Pr_{S,S',R}[\forall t \in [T] : a_t(S, R) = a_t(S', R)] \geq 1 - \rho.$$

## 3 Adversarially $\rho$-Replicable Online Linear Optimization

In this section, we present Algorithm 1 that is an adversarially $\rho$-replicable algorithm with sub-linear regret for online linear optimization. Our algorithm leverages two main ideas of first partitioning the time horizon into blocks and updating its action only at the endpoints of blocks, referred to as transition points, and second rounding of cumulative cost vectors to the grid points in a random grid. The rounding idea is inspired by the "Follow the Lazy Leader"(FLL) algorithm of Kalai and Vempala [2005]. However, we argue that their algorithm is not necessarily replicable and to fix this issue, we also need the idea of blocking timesteps and updating the algorithm's actions only at the transition points. Algorithm 1 is described in Section 3.1. We analyze the regret of Algorithm 1 by appealing to a different algorithm "Follow the Perturbed Leader with Block Updates" that incurs the same expected cost as FLL (Section D.1). Finally, Theorem 3.1 shows that by selecting the block size and noise level carefully Algorithm 1 achieves adversarial $\rho$-replicability with sublinear regret bounds.

### 3.1 Follow the Lazy Leader with Block Updates (FLLB($\varepsilon$))

Algorithm 1 is a modified version of the "Follow the Lazy Leader"(FLL) algorithm of Kalai and Vempala [2005]. FLL first picks a $n$-dimensional grid with spacing $1/\varepsilon$ where $\varepsilon$ is given as input, and a random offset $p \sim Unif([0, 1/\varepsilon)^n)$. Then at each iteration, it considers the cube $\sum_{i=1}^{t-1} c_i + [0, 1/\varepsilon)^n$, that is the cube starting at $\sum_{i=1}^{t-1} c_i$ with side length $1/\varepsilon$, and then the algorithm picks the unique grid point $g_{t-1}$ inside this cube (see Figure 1). Subsequently, it plays the action that minimizes the total cost pretending the cost vector is $g_{t-1}$, $a_t = \arg\min_{a \in \mathcal{A}} a \cdot g_{t-1}$. FLL achieves sublinear regret, however, in Example 3.1, we argue that FLL does not necessarily satisfy the adversarial $\rho$-replicability property.

**Example 3.1.** *Consider two different cost sequences $S, S'$ sampled from the same product distribution $\mathcal{P}$ arriving in an online fashion. Suppose there are only two actions $a_1, a_2 \in \mathcal{A}$ where $a_1$ has a lower cost in hindsight. Suppose $c_1 = c'_1 = (0, B)$, where $B < 0$. This means that day 0 gives a bonus to the second action and pretends that $a_2$ has a better performance initially. Therefore, the algorithm keeps predicting $a_2$ until after a sufficient number of examples arrive at timestep $\tau$ and it becomes clear that $a_1$ has better performance, and then, the algorithm shifts to outputting $a_1$ for the rest of the timesteps. However, since $S, S'$ are different sequences, it is not necessarily the case that the switching thresholds for $S$ and $S'$ are the same. Consequently, FLL is not necessarily adversarially $\rho$-replicable.*

In Algorithm 1, in addition to the rounding idea of FLL, we leverage the idea of *blocking* timesteps to overcome the challenge presented in Example 3.1. Since at each timestep $t$, both $S_t, S'_t$ are drawn from the same distribution $\mathcal{D}_t$, it is the case that the switching thresholds for $S$ and $S'$ are "approximately" the same. We use this intuition to *block* the timesteps and during each block output the same action. Now as long as the block sizes are large enough, the switching threshold for both sequences $S, S'$ would be within the same block. That helps us to achieve replicability. However, since we are outputting the same action for each block and not updating our decision at each round,

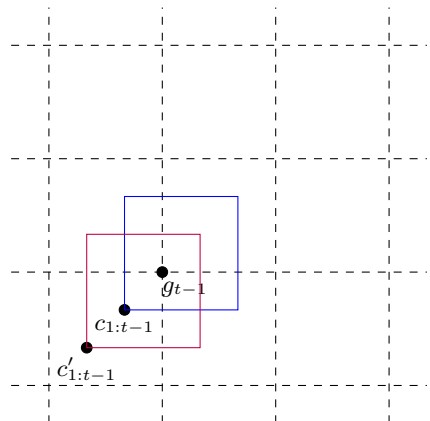

Figure 1: An illustration of the $\mathsf{FLLB}(\varepsilon, B)$ algorithm. The perturbed point $c_{1:t-1} + p$ is uniformly random over a cube of side $1/\varepsilon$ with vertex at $c_{1:t-1}$ (similarly for $c'_{t-1} + p$). In Theorem 3.1, we prove that by McDiarmid concentration bound, two different trajectories $c_{1:t-1}$ and $c'_{1:t-1}$ are within a distance $\Omega(\sqrt{nT})$ with high probability, and they get mapped to the same grid point $g_{t-1}$ with high probability.

the regret will suffer. Nevertheless, we show that by carefully selecting the block size, we can achieve adversarial $\rho$-replicability and sublinear regret guarantees.

Now we are ready to formally present our "Follow the Lazy Leader with Block Updates" algorithm (Algorithm 1). In the beginning, choose offset $p \in [0, \frac{1}{\varepsilon})^n$ uniformly at random, determining a grid $G = \{p + \frac{1}{\varepsilon}z \mid z \in \mathcal{Z}^n\}$, that is a grid that originates from $p$ with side length $1/\varepsilon$. Then at each iteration $t$, if $t - 1$ is a multiple of the block size $B$, the algorithm plays the action that minimizes the cost pretending the cost vector is $g_{t-1}$, where $g_{t-1}$ is the unique point in $G \cap \left(c_{1:t-1} + [0, \frac{1}{\varepsilon})^n\right)$, where $c_{1:t-1}$ is the cumulative cost from time 1 to $t-1$. Therefore, the action played by the algorithm in step $t$ is $a_t = \arg\min_{a \in \mathcal{A}} a \cdot g_{t-1}$. However, if $t - 1$ is not a multiple of $B$, the algorithm stays with the action played in the previous round. In Theorem 3.1, we prove that by setting the values of $\varepsilon$ and $B$ carefully, FLLB achieves strong replicability and sublinear regret guarantees.

---

**Algorithm 1** Follow the Lazy Leader with Block Updates ($\mathsf{FLLB}(\varepsilon, B)$)

---

**Input:** Sequence $S = \{c_1, \cdots, c_T\}$ arriving one by one over the time. $\varepsilon > 0$ and block size $B$.

1 Sample $p \sim [0, \frac{1}{\varepsilon})^n$ uniformly at random.
2 Let $a_1$ be a random action picked from $\mathcal{A}$.
3 Let $G \leftarrow \{p + \frac{1}{\varepsilon}z \mid z \in \mathcal{Z}^n\}$.
4 **for** $t : 1, \cdots, T$ **do**
5     **if** $t - 1$ *is a multiple of $B$* **then**
6         $g_{t-1} \leftarrow G \cap \left(c_{1:t-1} + [0, \frac{1}{\varepsilon})^n\right)$.
7         $a_t \leftarrow \arg\min_{a \in \mathcal{A}} a \cdot g_{t-1}$.
8     **else**
9         // Stay with the previous action.
        $a_t \leftarrow a_{t-1}$.

---

**Theorem 3.1** (Regret and Replicability Guarantees of FLLB). *$\mathsf{FLLB}(\varepsilon, B)$ is adversarially $\rho$-replicable and achieves regret $\widetilde{\mathcal{O}}(DT^{5/6}n^{1/6}\rho^{-1/3})$, when $B = \left(\frac{2(\sqrt{2\log(2T/\rho)}+2)\sqrt{nT}}{\rho}\right)^{2/3}$, and $\varepsilon = \frac{1}{\sqrt{BT}}$, and $D$ is the $\ell_1$ diameter of the action set $\mathcal{A}$.*

In order to prove the theorem, first, we analyze the regret of Algorithm 1. However, we need to be careful when picking the values of $\varepsilon$ and $B$ so that the regret does not blow up. If $\varepsilon$ is too large

then in Algorithm 1, a lot of different accumulated cost vectors $c_{1:t-1}$ get mapped to the same grid point $g_{t-1}$ which would cause a lot of regret. Similarly, when $B$ is too large, the algorithm takes the same action for a large block of time and does not update its decision which would cause the regret to suffer. In order to pick optimal values for $\varepsilon$ and $B$, we analyze the regret of Algorithm 1 by appealing to a different algorithm "Follow the Perturbed Leader with Block Updates" (FTPLB$(\varepsilon, B)$) that in expectation behaves identically to FLLB$(\varepsilon, B)$ on any single period and incurs the same expected cost. This algorithm is a modified version of the "Follow the Perturbed Leader" (FTPL$(\varepsilon)$) algorithm by Hannan-Kalai-Vempala.

### 3.2   Follow the Perturbed Leader with Block Updates (FTPLB)

First, we describe the FTPL$(\varepsilon)$ algorithm (Algorithm 5) by Hannan-Kalai-Vempala. FTPL$(\varepsilon)$ first hallucinates a fake day 0 with a random cost vector $c_0$, where $c_0 \sim Unif([0, \frac{1}{\varepsilon})^n)$. Then in each time period $t$, it picks the action that minimizes the cost of all days $c_0, \cdots, c_{t-1}$:

$$a_t = \arg\min_{a \in \mathcal{A}} \langle c_0 + c_1 + \cdots + c_{t-1}, a \rangle$$

---

**Algorithm 2** Follow the Perturbed Leader(FTPL$(\varepsilon)$) [Kalai and Vempala, 2005]

---

**Input:** Sequence $S = (c_1, \cdots, c_T)$ arriving one by one over the time, $\varepsilon > 0$.
10  Sample $c_0 \sim [0, \frac{1}{\varepsilon})^n$ uniformly at random.
11  **for** $t : 1, \cdots, T$ **do**
12  $\quad \lfloor \quad a_t \leftarrow \arg\min_{a \in \mathcal{A}} \sum_{i=0}^{t-1} a \cdot c_i$.

---

The same argument used in Example 3.1 demonstrates that Algorithm 5 is not adversarially $\rho$-replicable. Now, we describe the "Follow the Perturbed Leader with Block Updates"(FTPLB$(\varepsilon, B)$, Algorithm 6). FTPLB first hallucinates a fake day 0 with a random cost vector $c_0$, where $c_0 \sim [0, \frac{1}{\varepsilon})^n$ uniformly at random. Then, it picks the action that minimizes the cost of all days $c_0, \cdots, c_{B \lfloor t-1/B \rfloor}$, where $B$ is the block size.

$$a_t = \arg\min_{a \in \mathcal{A}} \langle c_0 + c_1 + \cdots + c_{B \lfloor (t-1)/B \rfloor}, a \rangle$$

---

**Algorithm 3** Follow the Perturbed Leader with Block Updates (FTPLB$(\varepsilon, B)$)

---

**Input:** Sequence $S = (c_1, \cdots, c_T)$ arriving one by one over the time. $\varepsilon > 0$ and block size $B$.
13  Sample $c_0 \sim [0, \frac{1}{\varepsilon})^n$ uniformly at random.
14  **for** $t : 1, \cdots, T$ **do**
15  $\quad$ **if** $t - 1$ *is a multiple of* $B$ **then**
16  $\quad\quad \lfloor \quad a_t \leftarrow \arg\min_{a \in \mathcal{A}} \sum_{i=0}^{t-1} a \cdot c_i$.
17  $\quad$ **else**
          $\quad\quad$ // Stay with the previous action.
18  $\quad\quad\quad a_t \leftarrow a_{t-1}$.
     $\quad\quad$ // this is equivalent to having $a_t = \arg\min_{a \in \mathcal{A}} \sum_{i=1}^{B \lfloor (t-1)/B \rfloor} a \cdot c_i$

---

**Remark 3.2.** *In each step $t$ when $t$ is a transition point,* FLLB *pretends that the cumulative cost vector is $g_{t-1}$ that is the unique grid point inside the cube $c_{1:t-1} + [0, 1/\varepsilon)^n$. Due to the random offset $p$ of the grid considered in* FLLB*, $g_{t-1}$ is uniformly distributed inside this cube. Furthermore,* FTPLB*, also pretends the cumulative cost vector is $c_{1:t-1} + c_0$ where $c_0 \sim Unif([0, 1/\varepsilon)^n)$, and therefore it has the same distribution as $g_{t-1}$. This implies,* FLLB *and* FTPLB *have the same expected cost.*

Next, we bound the regret of the FTPLB algorithm. The proof is deferred to the Appendix.

**Theorem 3.3** (Regret Guarantee of FTPLB). *Assume the maximum $\ell_1$ length of any cost vector $c \in \mathcal{C}$ is 1, and the $\ell_1$ diameter of the action set $\mathcal{A}$ is $D$. If $c_0 \sim Unif([0, 1/\varepsilon)^n)$ then the expected regret of* FTPLB$(\varepsilon, B)$ *in $T$ steps satisfies:*

$$\mathbb{E}[\mathsf{Reg}] \leq D(B\varepsilon T + 1/\varepsilon)$$

*setting $\varepsilon = 1/(\sqrt{BT})$ gives an expected regret at most $D\sqrt{BT}$.*

Finally, we are ready to provide a proof sketch of Theorem 3.1. The full proof is deferred to Section D.

*Proofsketch of Theorem 3.1.* First, we bound the probability of non-replicability. Recall that Algorithm 1 chooses a uniformly random grid of spacing $1/\varepsilon$. In each transition $t$, if $t$ is a transition point $\{B+1, 2B+1, \cdots, \}$, Algorithm 1 picks the action that minimizes the total cost, pretending the cumulative cost vector is $g_{t-1}$, where $g_{t-1}$ is a grid point that lies inside $c_{1:t-1} + [0, 1/\varepsilon)^n$ (see Figure 1). Consider two different trajectories $c_1, \cdots, c_t$ and $c'_1, \cdots, c'_t$ where each $c_i, c'_i \sim \mathcal{D}_i$. First, we bound the probability that $g_{t-1} \neq g'_{t-1}$ as follows. This event happens iff the grid point in $c_{1:t-1} + [0, \frac{1}{\varepsilon}]^n$ is not in $c'_{1:t-1} + [0, \frac{1}{\varepsilon}]^n$. In the proof, we argue that for any $v \in \mathbb{R}^n$, the cubes $[0, 1/\varepsilon]^n$ and $v + [0, 1/\varepsilon]^n$ overlap in at least a $(1 - \varepsilon \ell_1(v))$ fraction. Furthermore, we prove that with high probability, the $\ell_1$ distance between cumulative cost vectors $c_{1:t}$ and $c'_{1:t}$ is bounded as follows when $p = \sqrt{2 \log(2T/\rho)} + 2$.

$$\Pr\left(\ell_1(c_{1:t} - c'_{1:t}) \geq p\sqrt{nT}\right) \leq (\frac{\rho}{2T})^n \leq \frac{\rho}{2T}$$

Now by taking union bound over all the transition points $t = \{B+1, 2B+1, \cdots\}$ the total probability of non-replicability is at most $(\frac{\rho}{2T} + \varepsilon p\sqrt{nT})\frac{T}{B}$. Consequently, we can set values of $B, \varepsilon$ such that the probability of non-replicability over all the time-steps is at most $\rho$:

$$(\frac{\rho}{2T} + \varepsilon p\sqrt{nT})\frac{T}{B} \leq \rho$$

to achieve this, it suffices to have:

$$\varepsilon = \frac{B\rho}{2p\sqrt{n}T^{3/2}}$$

Additionally, in order to minimize regret by Theorem 3.3, we need to set $\varepsilon = \frac{1}{\sqrt{BT}}$. In the proof we show that by setting $B = \widetilde{O}(T^{2/3}n^{1/3}(1/\rho)^{2/3})$, we get a regret bound of $\mathcal{O}(D\sqrt{BT}) = \widetilde{\mathcal{O}}(DT^{5/6}n^{1/6}\rho^{-1/3})$. $\qquad\square$

## 4 Experts

In this section, we investigate the experts problem and present our algorithm "Follow the Perturbed Leader with Block Updates and Geometric Noise"(FTPLB$^\star$) (Algorithm 4) that is an adversarially $\rho$-replicable learning algorithm with sublinear regret. This algorithm applies the strategy of partitioning the time horizon into blocks and only updating its actions at the end of each block. Initially, geometric noise is added to each expert, and at the end of each block, the algorithm chooses the expert with the lowest cumulative cost, considering the initial noise. Theorem 4.1 demonstrates that with appropriately chosen block sizes and noise levels, it is possible to achieve no-regret learning along with adversarial $\rho$-replicability.

---

**Algorithm 4** Follow the Perturbed Leader with Geometric Noise and Block Updates (FTPLB$^\star(\varepsilon, B)$)

---

**Input:** Sequence $S = (c_1, \cdots, c_T)$ arriving one by one over the time. $\varepsilon > 0$ and block size $B$.

19 Sample for every expert $a \in \mathcal{A}$, $X_a \sim \mathbf{Geo}(\varepsilon)$.

20 **for** $t : 1, \cdots, T$ **do**

21     **if** $t - 1$ *is a multiple of* $B$ **then**

22         $a_t \leftarrow \arg\min_{a \in \mathcal{A}} \sum_{i=1}^{t-1} c_i(a) - X_a$.

23     **else**

        // Stay with the previous action.

24         $a_t \leftarrow a_{t-1}$.

    // this is equivalent to having $a_t = \arg\min_{a \in \mathcal{A}} \sum_{i=1}^{B\lfloor (t-1)/B \rfloor} c_i(a)$

---

**Theorem 4.1** (Regret and Replicability Guarantees of FTPLB$^\star$). FTPLB$^\star(\varepsilon, B)$ *is adversarially $\rho$-replicable and achieves regret $\widetilde{\mathcal{O}}(T^{5/6} \ln^{5/6}(n)\rho^{-1/3})$ when $B = \left(\frac{8\sqrt{2(\frac{\log(8T/\rho)}{\log n}+1)}\ln(n)T}{\rho}\right)^{2/3}$ and $\varepsilon = \sqrt{\frac{\ln n}{BT}}$.*

Proof is deferred to Section E.

## 5 A General Framework for Replicable Online Learning

In this section, we demonstrate a general recipe for converting a regret minimization algorithm $\mathsf{ALG}_{int}$, which we call the internal algorithm, to an adversarially $\rho$-replicable algorithm $\mathsf{ALG}_{ext}$ with potentially a higher regret. Our general procedure uses two main ideas, a rounding procedure of the loss trajectories inspired by the "Follow the Lazy Leader"(FLL) algorithm of Kalai and Vempala [2005], and blocking of time-steps. Here we present the results for the setting where the internal algorithm $\mathsf{ALG}_{int}$ takes as input bounded $\ell_1$ cost vectors. Section G provides the results for the $\ell_\infty$ case. As mentioned in the introduction, we require the further assumption that the cost function $\mathsf{Cost}(a, c)$ is linear in $c$, i.e., $\mathsf{Cost}(a, c_1 + c_2) = \mathsf{Cost}(a, c_1) + \mathsf{Cost}(a, c_2)$. However, this is still sufficient to capture both the settings of online linear optimization and experts problem.

Our framework is as follows: At each timestep $t$ if $t - 1$ is a multiple of the block size $B$, then, it picks an $n$-dimensional grid with spacing $1/\varepsilon$ where $\varepsilon$ is given as input, and a random offset $p_t \sim [0, 1/\varepsilon)^n$, where $n$ is the dimension of the space. Then it considers the cube $c_{1:t-1} + [0, 1/\varepsilon)^n$, where $c_{1:k} = \sum_{i=1}^{k} c_i$, and picks the unique grid points $g_{t-1}$ inside this cube. Furthermore, for a fresh random offset $p_t' \sim [0, 1/\varepsilon)^n$, it picks a new $n$-dimensional grid with spacing $1/\varepsilon$ and offset $p_t'$. Then, it considers the cube $c_{1:t-1-B} + [0, 1/\varepsilon)^n$ and picks the unique grid point $g_{t-1-B}'$ inside this cube. Then, it gives $\frac{1}{B+2n/\varepsilon}(g_{t-1} - g_{t-1-B}')$ as the input argument to the algorithm $\mathsf{ALG}_{int}$, and plays the action returned by $\mathsf{ALG}_{int}$ for the next $B$ timesteps. This framework is given in Algorithm 7. Theorem 5.1 demonstrates that Algorithm 7 is adversarially $\rho$-replicable with bounded regret.

**Theorem 5.1** (Regret and Replicability Guarantees of Algorithm 7). *Let $\rho > 0$ be a parameter, and assume that the cost function $\mathsf{Cost}(a, c)$ is linear in $c$, i.e., $\mathsf{Cost}(a, c_1 + c_2) = \mathsf{Cost}(a, c_1) + \mathsf{Cost}(a, c_2)$.*

*Suppose for two different input trajectories $c_1, \cdots, c_T$ and $d_1, \cdots, d_T$ where for each time-step $t$, $c_t, d_t \sim \mathcal{D}_i$, for each time step $t$, the $\ell_1$ distance between $c_{1:t}$ and $d_{1:t}$ is at most $m$ with probability at least $1 - \gamma$. Then, Algorithm 7 is adversarially $\rho$-replicable and achieves cumulative regret $\mathcal{O}\left(B\mathsf{Reg}_{T/B}(\mathsf{ALG}_{int})\right)$ when $\gamma$ is at most $\frac{\rho B}{4T}$, $B = \sqrt{\frac{8nmT}{\rho}}$ and $\varepsilon = 2n/B$.*

Proof is deferred to Section F. Next, we show the application of Algorithm 7 to the online linear optimization problem.

**Corollary 5.2.** *In the online linear optimization setting, Algorithm 7 is adversarially $\rho$-replicable and achieves cumulative regret $\widetilde{\mathcal{O}}(DT^{7/8}n^{3/8}\rho^{-1/4})$.*

## 6 Further Directions & Open Quesitons

The questions we address in this paper fall within the full-information setting, specifically the experts problem and online linear optimization. A natural future direction is to extend these techniques to the bandit/partial-information setting while requiring *adversarial replicability*, where we believe our ideas could be valuable.

As mentioned, it would also be interesting to explore other problems, such as clustering and reinforcement learning, that have been studied in the iid context, and extend them to the adversarial replicability setting.

Additionally, the current lower bounds on regret (Theorems I.2 and I.3), while optimal for the iid-replicability setting, do not match the upper bounds in the adversarial replicability setting for either the experts problem or online linear optimization. An intriguing open question is to prove an $\omega(\sqrt{T})$ lower bound on regret for a fixed value of $\rho$, say $1/100$, in the experts setting, or to design an algorithm that achieves $O(\sqrt{T})$ regret while remaining $\rho$-replicable.

**Acknowledgements**

This work was supported in part by the National Science Foundation under grants CCF-2212968 and ECCS-2216899, by the Simons Foundation under the Simons Collaboration on the Theory of Algorithmic Fairness, and by the Office of Naval Research MURI Grant N000142412742. The views expressed in this work do not necessarily reflect the position or the policy of the Government and no official endorsement should be inferred. The research was done when SA and SB were at Toyota Technological Institute at Chicago.

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

## A  Further Related Work

Online learning is a well-studied and classical field, with extensive literature on the subject. Our work is inspired by Kalai and Vempala [2005].

Algorithmic replicability was independently introduced by Impagliazzo et al. [2022], Ghazi et al. [2021], Ahn et al. [2024], and later studied in the context of multi-armed bandits Esfandiari et al. [2022], reinforcement learning Eaton et al. [2024], clustering Esfandiari et al. [2024]. Several works have shown statistical equivalences and separations between replicability and other notions of algorithmic stability including differential privacy [Bun et al., 2023, Kalavasis et al., 2023]. In fact, replicable bandit algorithms have also found use in medical contexts Zhang et al. [2024]. However, all of these works focus on the *iid-replicability setting*, making it intriguing to explore these problems in the *adversarial replicability setting*.

## B  Some Standard Concentration Inequalities

**Theorem B.1** (Blackwell [1997](Theorem 1) (see Howard et al. [2020]))**.** *Let $S_t$ be a real valued martingale with $S_0 = 0$. If $|\Delta S_t| \leq 1$ for all t, then for any $a, b > 0$, we have*

$$\mathbb{P}\left(\exists t \in \mathbb{N} : S_t \geq a + bt\right) \leq e^{-2ab}.$$

*Therefore, we also have*

$$\mathbb{P}\left(\exists t \in \mathbb{N} : S_t \leq -a - bt\right) \leq e^{-2ab}.$$

**Theorem B.2** (McDiarmid's Inequality McDiarmid [1989])**.** *Let $X_1, X_2, \ldots, X_n$ be independent random variables taking values in a set $\mathcal{X}$, and let $f : \mathcal{X}^n \to \mathbb{R}$ be a function such that for all $i$ and all $x_1, \ldots, x_n, x_i' \in \mathcal{X}$,*

$$|f(x_1, \ldots, x_i, \ldots, x_n) - f(x_1, \ldots, x_i', \ldots, x_n)| \leq c_i,$$

*for some constants $c_1, c_2, \ldots, c_n$. Then, for any $\varepsilon > 0$,*

$$\mathbb{P}\left(f(X_1, X_2, \ldots, X_n) - \mathbb{E}[f(X_1, X_2, \ldots, X_n)] \geq \varepsilon\right) \leq \exp\left(-\frac{2\varepsilon^2}{\sum_{i=1}^n c_i^2}\right),$$

*and*

$$\mathbb{P}\left(f(X_1, X_2, \ldots, X_n) - \mathbb{E}[f(X_1, X_2, \ldots, X_n)] \leq -\varepsilon\right) \leq \exp\left(-\frac{2\varepsilon^2}{\sum_{i=1}^n c_i^2}\right).$$

## C  Concentration of trajectories in $\ell_1$-norm

**Lemma C.1** (Concentration of trajectories in $\ell_1$-norm)**.** *Let $\mathcal{D}_1, \ldots, \mathcal{D}_t$ be distributions over the unit $\ell_1$ ball in $\mathbb{R}^t$. Consider two independent sequences of independent random $n$-dimensional vectors $\{v_i\}_{i=1}^t$ and $\{w_i\}_{i=1}^t$, where each $v_i$ and $w_i$ is drawn from $\mathcal{D}_i$. Then, for $c > 2$ we have*

$$\Pr\left[\left|\sum_{i=1}^t (v_i - w_i)\right|_1 > c\sqrt{tn}\right] \leqslant \exp\left(-\frac{(c-2)^2 n}{2}\right).$$

*Proof.* Let $u_i = v_i - w_i$. Note that always $\|u_i\|_1 \leqslant 2$. Let $f(u_1, \ldots, u_t) = \|\Sigma_i^t u_i\|_1$ Then $f(u_1, \ldots, u_{i-1}, -, u_{i+1}, \ldots, u_t) : \mathbb{R} \to \mathbb{R}$ is 2-Lipschitz for all $i \in [t]$. Hence by McDiarmid's inequality (see Theorem B.2),

$$\Pr\left[f(u_1, \ldots, u_t) - E[f(u_1, \ldots, u_t)] > \varepsilon\right] \leqslant \exp\left(\frac{-2\varepsilon^2}{4t}\right).$$

Now, for a vector $u$ let $u^{(k)}$ denote the $k^{th}$ component. Then,

$$\mathbb{E}[f(u_1, \ldots, u_t)] = \mathbb{E}[\|\Sigma_i u_i\|_1]$$
$$= \mathbb{E}\left[\sum_{k=1}^n \left|\sum_{i=1}^t u_i^{(k)}\right|\right]$$
$$= \sum_{k=1}^n \mathbb{E}\left[\left|\sum_{i=1}^t u_i^{(k)}\right|\right].$$

Applying Jensen's inequality we have:

$$\sum_{k=1}^{n} \mathbb{E}\left[\left|\sum_{i=1}^{t} u_i^{(k)}\right|\right] \leqslant \sum_{k=1}^{n} \sqrt{\mathbb{E}\left[\left(\sum_{i} u_i^{(k)}\right)^2\right]}$$

which by canceling cross terms gives

$$= \sum_{k=1}^{n} \sqrt{\mathbb{E}\left[\sum_{i} \left(u_i^{(k)}\right)^2\right]}.$$

Further, by the CauchySchwarz inequality we have:

$$\sum_{k=1}^{n} \sqrt{\mathbb{E}\left[\sum_{i} \left(u_i^{(k)}\right)^2\right]} = \left\langle \mathbf{1}^n, \left(\sqrt{\mathbb{E}\left[\sum_{i} \left(u_i^{(1)}\right)^2\right]}, \ldots, \sqrt{\mathbb{E}\left[\sum_{i} \left(u_i^{(n)}\right)^2\right]}\right)\right\rangle$$

$$\leqslant \sqrt{n} \times \sqrt{\sum_{k=1}^{n} \mathbb{E}\left[\sum_{i=1}^{t} \left(u_i^{(k)}\right)^2\right]}$$

$$= \sqrt{n} \times \sqrt{\mathbb{E}\left[\sum_{k=1}^{n}\sum_{i=1}^{t} \left(u_i^{(k)}\right)^2\right]}$$

$$= \sqrt{n} \times \sqrt{\mathbb{E}\left[\sum_{i=1}^{t} \|u_i\|_2^2\right]}$$

$$= 2\sqrt{tn} \qquad (|u_i|_2 \leqslant 2 \text{ as } |u_i|_1 \leqslant 2)$$

$$\Rightarrow \mathbb{E}\left[f\left(u_1, \ldots, u_i\right)\right] \leqslant 2\sqrt{tn}.$$

Finally, setting $\varepsilon = (c-2)\sqrt{tn}$ we have,

$$\Pr\left[f\left(u_1, \ldots, u_t\right) > c\sqrt{tn}\right] \leqslant \exp\left(-\frac{(c-2)^2 n}{2}\right),$$

which proves the lemma. $\qquad\square$

## D  Proof of Theorem 3.1

First, we analyze the regret of Algorithm 1. However, we need to be careful when picking the values of $\varepsilon$ and $B$ so that the regret does not blow up. If $\varepsilon$ is too large then in Algorithm 1, a lot of different accumulated cost vectors $c_{1:t-1}$ get mapped to the same grid point $g_{t-1}$ which would cause a lot of regret. Similarly, when $B$ is too large, the algorithm takes the same action for a large block of time and does not update its decision which would cause the regret to suffer. In order to pick optimal values for $\varepsilon$ and $B$, we analyze the regret of Algorithm 1 by appealing to a different algorithm "Follow the Perturbed Leader with Block Updates" (FTPLB$(\varepsilon, B)$) that in expectation behaves identically to FLLB$(\varepsilon, B)$ on any single period and incurs the same expected cost. This algorithm is a modified version of the "Follow the Perturbed Leader" (FTPL$(\varepsilon)$) algorithm by Hannan-Kalai-Vempala.

### D.1  Follow the Perturbed Leader with Block Updates (FTPLB)

First, we describe the FTPL$(\varepsilon)$ algorithm (Algorithm 5) by Hannan-Kalai-Vempala. FTPL$(\varepsilon)$ first hallucinates a fake day 0 with a random cost vector $c_0$, where $c_0 \sim Unif([0, \frac{1}{\varepsilon})^n)$. Then in each time period $t$, it picks the action that minimizes the cost of all days $c_0, \cdots, c_{t-1}$:

$$a_t = \arg\min_{a \in \mathcal{A}} \langle c_0 + c_1 + \cdots + c_{t-1}, a \rangle$$

---
**Algorithm 5** Follow the Perturbed Leader(FTPL($\varepsilon$)) [Kalai and Vempala, 2005]

---
**Input:** Sequence $S = (c_1, \cdots, c_T)$ arriving one by one over the time, $\varepsilon > 0$.

25 Sample $c_0 \sim [0, \frac{1}{\varepsilon})^n$ uniformly at random.

26 **for** $t : 1, \cdots, T$ **do**

27 $\quad \lfloor \quad a_t \leftarrow \arg\min_{a \in \mathcal{A}} \sum_{i=0}^{t-1} a \cdot c_i.$

---

The same argument used in Example 3.1 demonstrates that Algorithm 5 is not adversarially $\rho$-replicable. Now, we describe the "Follow the Perturbed Leader with Block Updates"(FTPLB($\varepsilon$, $B$), Algorithm 6). FTPLB first hallucinates a fake day 0 with a random cost vector $c_0$, where $c_0 \sim [0, \frac{1}{\varepsilon})^n$ uniformly at random. Then, it picks the action that minimizes the cost of all days $c_0, \cdots, c_{B\lfloor t-1/B \rfloor}$, where $B$ is the block size.

$$a_t = \arg\min_{a \in \mathcal{A}} \langle c_0 + c_1 + \cdots + c_{B\lfloor (t-1)/B \rfloor}, a \rangle$$

---
**Algorithm 6** Follow the Perturbed Leader with Block Updates (FTPLB($\varepsilon$, $B$))

---
**Input:** Sequence $S = (c_1, \cdots, c_T)$ arriving one by one over the time. $\varepsilon > 0$ and block size $B$.

28 Sample $c_0 \sim [0, \frac{1}{\varepsilon})^n$ uniformly at random.

29 **for** $t : 1, \cdots, T$ **do**

30 $\quad$ **if** $t - 1$ *is a multiple of $B$* **then**

31 $\quad\quad \lfloor \quad a_t \leftarrow \arg\min_{a \in \mathcal{A}} \sum_{i=0}^{t-1} a \cdot c_i.$

32 $\quad$ **else**

$\quad\quad$ // Stay with the previous action.

33 $\quad\quad \lfloor \quad a_t \leftarrow a_{t-1}.$

$\quad$ // this is equivalent to having $a_t = \arg\min_{a \in \mathcal{A}} \sum_{i=1}^{B\lfloor (t-1)/B \rfloor} a \cdot c_i$

---

**Remark D.1.** *In each step $t$ when $t$ is a transition point,* FLLB *pretends that the cumulative cost vector is $g_{t-1}$ that is the unique grid point inside the cube $c_{1:t-1} + [0, 1/\varepsilon)^n$. Due to the random offset $p$ of the grid considered in* FLLB*, $g_{t-1}$ is uniformly distributed inside this cube. Furthermore,* FTPLB*, also pretends the cumulative cost vector is $c_{1:t-1} + c_0$ where $c_0 \sim Unif([0, 1/\varepsilon)^n)$, and therefore it has the same distribution as $g_{t-1}$. This implies,* FLLB *and* FTPLB *have the same expected cost.*

In order to bound the regret of Algorithm 6, first we show the following lemma holds.

**Lemma D.2.** *First, we show that the non-implementable[2] "Be the Leader" (BTL) algorithm of picking*

$$a_t = \arg\min_{a \in \mathcal{A}} \langle c_1 + \cdots + c_t, a \rangle$$

*has zero (or negative) regret:*

$$\langle c_1, a_1 \rangle + \langle c_2, a_2 \rangle + \cdots + \langle c_T, a_T \rangle \leq \langle c_1, a_T \rangle + \langle c_2, a_T \rangle + \cdots + \langle c_T, a_T \rangle.$$

*Proof.* We prove this by induction on $T$.

**Base case:** For $T = 1$, the left-hand side and right-hand side are equal.

**General case:** By induction, we can assume:

$$\langle c_1, a_1 \rangle + \langle c_2, a_2 \rangle + \cdots + \langle c_{T-1}, a_{T-1} \rangle \leq \langle c_1 + c_2 + \cdots, c_{T-1}, a_{T-1} \rangle.$$

We also know by definition of $a_{T-1}$, that:

$$\langle c_1 + c_2 + \cdots, c_{T-1}, a_{T-1} \rangle \leq \langle c_1 + \cdots + c_{T-1}, a_T \rangle.$$

Putting these together, and adding $\langle c_T, a_T \rangle$ to both sides, yields the theorem.

$\square$

---
[2] it is not implementable since it assumes at time $t$ the algorithm knows the cost vector $c_t$ before picking action $a_t$.

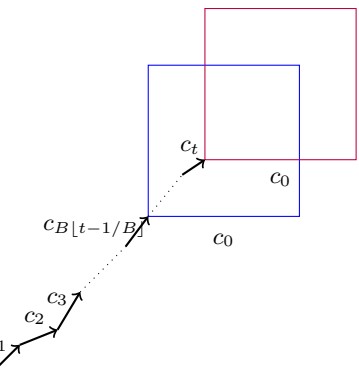

Figure 2: An illustration of the FTPLB algorithm. The blue and red boxes are corresponding to FTPLB and BTPL respectively.

**Theorem 3.3** (Regret Guarantee of FTPLB). *Assume the maximum $\ell_1$ length of any cost vector $c \in \mathcal{C}$ is 1, and the $\ell_1$ diameter of the action set $\mathcal{A}$ is $D$. If $c_0 \sim \text{Unif}([0, 1/\varepsilon]^n)$ then the expected regret of $\text{FTPLB}(\varepsilon, B)$ in $T$ steps satisfies:*

$$\mathbb{E}[\text{Reg}] \leq D(B\varepsilon T + 1/\varepsilon)$$

*setting $\varepsilon = 1/(\sqrt{BT})$ gives an expected regret at most $D\sqrt{BT}$.*

*Proof of Theorem 3.3.* We consider another non-implementable algorithm BTPL and argue that the expected cost paid by FTPLB at time $t$ is close to the expected cost paid by BTPL at time $t$. BTPL takes the action $a_t = \arg\min_{a \in \mathcal{A}} \sum_{i=0}^{t} a \cdot c_i$. Here, similar to BTL, BTPL assumes knowledge of cost vector $c_t$ before choosing $a_t$ which is infeasible. Moreover, we demonstrate that the expected cost of BTPL is close to BTL, and since BTL has zero (or negative) regret, we can upper bound the regret of FTPL.

FTPLB is choosing its objective function $a_t = \arg\min_{a \in \mathcal{A}} \sum_{i=1}^{B\lfloor (t-1)/B \rfloor} a \cdot c_i$ uniformly at random from a box of side-length $1/\varepsilon$ with corner at $c_1 + \cdots + c_{B\lfloor t-1/B \rfloor}$. BTPL is choosing its objective function uniformly at random from a box of side-length $1/\varepsilon$ with corner at $c_1 + \cdots + c_t$ (Figure 2). We call these boxes $\text{Box}^{\text{FTPLB}}$ and $\text{Box}^{\text{BTPL}}$ respectively. These boxes each have volume $V = (1/\varepsilon)^n$. Their bottom corners differ by the sum of vector $c_{B\lfloor t-1/B \rfloor + 1}, \cdots, c_t$ each of $\ell_1$ length at most 1, and therefore in total differing by an $\ell_1$ length at most $B$. As a result, the intersection of the boxes has volume at least $(1/\varepsilon)^n - B(1/\varepsilon)^{n-1} \geq V(1 - B\varepsilon)$.

This implies that for some $p \leq B\varepsilon$, we can view FTPLB as with probability $1 - p$ choosing its objective function uniformly at random from $I = \text{Box}_t^{\text{FTPLB}} \cap \text{Box}_t^{\text{BTPL}}$, and with probability $p$ choosing its objective function uniformly at random from $F = \text{Box}_t^{\text{FTPLB}} \setminus \text{Box}_t^{\text{BTPL}}$. Similarly, we can view BTPL as with probability $1 - p$ choosing its objective uniformly at random from $I$, and with probability $p$ choosing its objective function uniformly at random from $B = \text{Box}_t^{\text{BTPL}} \setminus \text{Box}_t^{\text{FTPLB}}$. Thus, if $\alpha_I = \mathbb{E}_{c \sim I}[\langle c_t, a_t \rangle \mid a_t = \arg\min_{a \in \mathcal{A}} \langle c, a \rangle]$, $\alpha_F = \mathbb{E}_{c \sim F}[\langle c_t, a_t \rangle \mid a_t = \arg\min_{a \in \mathcal{A}} \langle c, a \rangle]$, and $\alpha_B = \mathbb{E}_{c \sim B}[\langle c_t, a_t \rangle \mid a_t = \arg\min_{a \in \mathcal{A}} \langle c, a \rangle]$, then the expected cost of FTPLB at time $t$ is $(1 - p)\alpha_I + p\alpha_F$ and the expected cost of BTPL at time $t$ is $(1 - p)\alpha_I + p\alpha_B$.

Now, if we prove that $|\alpha_F - \alpha_B| \leq D$, then the difference in expected cost between FTPL and BTPL at time $t$ is at most $p|\alpha_F - \alpha_B| \leq BD\varepsilon$. Summing over all times $t$ we get a difference at most $BDT\varepsilon$. To prove that $|\alpha_F - \alpha_B| \leq D$, note that for any $c \in \mathcal{C}$ and $a, a' \in \mathcal{A}$, we have $\langle c, a \rangle - \langle c, a' \rangle \leq D$. This follows from the fact that the $\ell_\infty$ length of $c$ is at most 1 (this follows from the assumption that $\ell_1$ length of $c$ is at most 1) and the $\ell_1$ diameter of $\mathcal{A}$ is $D$. Consequently, $|\alpha_F - \alpha_B| \leq D$.

Next, we bound the difference between the cost of BTPL and BTL. Let $a_t^{\text{BTPL}}$ and $a_t^{\text{BTL}}$ show the actions taken by BTPL and BTL at time $t$ respectively. Let $a_T^{\text{BTL}}$ be the optimal action in hindsight. For any vector $c_0$, by Lemma D.2, we have

$$\langle c_0, a_0^{\text{BTPL}} \rangle + \cdots + \langle c_T, a_T^{\text{BTPL}} \rangle \leq \langle (c_0 + \cdots + c_T), a_T^{\text{BTPL}} \rangle \leq \langle (c_0 + \cdots + c_T), a_T^{\text{BTL}} \rangle.$$

Therefore,

$$\langle c_1, a_1^{\text{BTPL}} \rangle + \cdots + \langle c_T, a_T^{\text{BTPL}} \rangle \le \langle (c_1 + \cdots + c_T), a_T^{\text{BTL}} \rangle + \langle c_0, a_T^{\text{BTL}} - a_0^{\text{BTPL}} \rangle. \tag{1}$$

The left-hand side of Equation (1) is the cost of BTPL. The first term on the right-hand side of Equation (1) is the cost of the optimal action in hindsight. Therefore, the expected regret of BTPL is at most $\mathbb{E}[\langle c_0, a_T^{\text{BTL}} - a_0^{\text{BTPL}} \rangle]$. Since $\mathcal{A}$ has $\ell_1$ diameter at most $D$, and each coordinate of $c_0$ has expected value $1/\varepsilon$, therefore $\mathbb{E}[\langle c_0, a_T^{\text{BTL}} - a_0^{\text{BTPL}} \rangle] \le D/\varepsilon$.

This implies the regret of FTPL is at most $BDT\varepsilon + D/\varepsilon$ and the proof is complete. $\qquad\square$

## D.2 Proof of Theorem 3.1

Now we are ready to analyze regret and replicability guarantees of FLLB($\varepsilon$).

**Theorem 3.1** (Regret and Replicability Guarantees of FLLB). *FLLB($\varepsilon, B$) is adversarially $\rho$-replicable and achieves regret $\widetilde{\mathcal{O}}(DT^{5/6}n^{1/6}\rho^{-1/3})$, when $B = \left( \frac{2(\sqrt{2\log(2T/\rho)}+2)\sqrt{nT}}{\rho} \right)^{2/3}$, and $\varepsilon = \frac{1}{\sqrt{BT}}$, and $D$ is the $\ell_1$ diameter of the action set $\mathcal{A}$.*

*Proof.* First, we bound the probability of non-replicability. Recall that Algorithm 1 chooses a uniformly random grid of spacing $1/\varepsilon$. In each transition $t$, if $t$ is a transition point $\{B+1, 2B+1, \cdots, \}$, Algorithm 1 picks the action that minimizes the total cost, pretending the cumulative cost vector is $g_{t-1}$, where $g_{t-1}$ is a grid point that lies inside $c_{1:t-1} + [0, 1/\varepsilon)^n$ (see Figure 1). Consider two different trajectories $c_1, \cdots, c_t$ and $c'_1, \cdots, c'_t$ where each $c_i, c'_i \sim \mathcal{D}_i$. First, we bound the probability that $g_{t-1} \ne g'_{t-1}$ as follows. This event happens iff the grid point in $c_{1:t-1} + [0, \frac{1}{\varepsilon}]^n$ is not in $c'_{1:t-1} + [0, \frac{1}{\varepsilon}]^n$. First, we argue that for any $v \in \mathbb{R}^n$, the cubes $[0, 1/\varepsilon]^n$ and $v + [0, 1/\varepsilon]^n$ overlap in at least a $(1 - \varepsilon \ell_1(v))$ fraction. Take a random point $x \in [0, 1/\varepsilon]^n$. If $x \notin v + [0, 1/\varepsilon]^n$, then for some $i$, $x_i \notin v_i + [0, 1/\varepsilon]$ which happens with probability at most $\varepsilon \ell_1(v_i)$ for any particular $i$. By the union bound, we are done. Thus in order to upper bound the probability that $g_{t-1} \ne g'_{t-1}$, we need to upper bound the $\ell_1$ distance between cumulative cost vectors $c_{1:t}$ and $c'_{1:t}$. For any $1 \le t \le T$:

$$\Pr\left( \ell_1(c_{1:t} - c'_{1:t}) \ge p\sqrt{nT} \right) \tag{2}$$

$$\le \Pr\left( \ell_1(c_{1:t} - c'_{1:t}) \ge p\sqrt{nt} \right) \le \exp\left(-\frac{(p-2)^2 n}{2}\right) \tag{3}$$

$$\le \frac{1}{\exp(n\log(2T/\rho))} \le \left(\frac{\rho}{2T}\right)^n \le \frac{\rho}{2T} \tag{4}$$

where Equation (3) holds by Lemma C.1, and Equation (4) holds by setting $p = \sqrt{2\log(2T/\rho)} + 2$. This implies that the probability of non-replicability at a transition point $t$, i.e. $g_{t-1} \ne g'_{t-1}$, is at most $\frac{\rho}{2T} + \varepsilon p\sqrt{nT}$. By taking union bound over all the transition points $t = \{B+1, 2B+1, \cdots\}$ the total probability of non-replicability is at most $(\frac{\rho}{2T} + \varepsilon p\sqrt{nT})\frac{T}{B}$. Consequently, we can set values of $B, \varepsilon$ such that the probability of non-replicability over all the time-steps is at most $\rho$:

$$\left(\frac{\rho}{2T} + \varepsilon p\sqrt{nT}\right)\frac{T}{B} \le \rho$$

in order to achieve this, it suffices to have:

$$\frac{\varepsilon p\sqrt{n}T^{3/2}}{B} = \rho/2$$

or equivalently:

$$\varepsilon = \frac{B\rho}{2p\sqrt{n}T^{3/2}}$$

Additionally, in order to minimize regret by Theorem 3.3, we need to set $\varepsilon = \frac{1}{\sqrt{BT}}$. Now by setting

$$\frac{B\rho}{2p\sqrt{n}T^{3/2}} = \frac{1}{\sqrt{BT}}$$

it implies that

$$B = \left(\frac{2(\sqrt{2\log(2T/\rho)} + 2)\sqrt{n}T}{\rho}\right)^{2/3} = \widetilde{O}(T^{2/3}n^{1/3}(1/\rho)^{2/3})$$

Finally, plugging in the value of $B$ in Theorem 3.3, gives a regret bound of $\mathcal{O}(D\sqrt{BT}) = \widetilde{\mathcal{O}}(DT^{5/6}n^{1/6}\rho^{-1/3})$. $\qquad\square$

## E  Missing Proofs of Section 4 (Proof of Theorem 4.1)

In order to prove Theorem 4.1, first, we bound the regret of $\mathsf{FTPLB}^\star(\varepsilon, B)$.

**Theorem E.1.** *The expected regret of* $\mathsf{FTPLB}^\star(\varepsilon, B)$ *in $T$ steps satisfies:*

$$\mathbb{E}[\mathsf{Reg}(\mathsf{FTPLB}^\star)] \le \varepsilon BT + \frac{\ln n}{\varepsilon}$$

*setting* $\varepsilon = \sqrt{\frac{\ln n}{BT}}$ *gives an expected regret at most* $\sqrt{BT\ln n}$.

In order to prove Theorem E.1, first, consider the non-implementable algorithm "Be the Perturbed Leader" (BTPL) that takes action $a_t^{\mathsf{BTPL}} = \arg\min_{a\in\mathcal{A}} \sum_{i=1}^{t} c_i(a) - X_a$ in each round has small regret. It is non-implementable since it assumes knowledge of the cost vector of the current step before taking an action in the current period.

**Lemma E.2.** *For any expert* $a \in \mathcal{A}$,

$$\mathbb{E}[\mathsf{Cost}(\mathsf{BTPL})] \le \mathsf{Cost}(a) + \mathbb{E}[\max_{a\in\mathcal{A}} X_a]$$

*Proof.* We assume a time zero and assume $c_0(a) = -X_a$ for simplicity of notation. Then it is the case that:

$$a_t^{\mathsf{BTPL}} = \arg\min_{a\in\mathcal{A}} \sum_{i=0}^{t} c_i(a)$$

Let $a_t^{\mathsf{BTL}}$ be the action taken by the non-implementable "Be the Leader" algorithm that is similar to BTPL but does not take into account the noise on each expert:

$$a_t^{\mathsf{BTL}} = \arg\min_{a\in\mathcal{A}} \sum_{i=1}^{t} c_i(a)$$

Then $a_T^{\mathsf{BTL}}$ is the optimal action in hindsight given the true cost sequence $c_1, \cdots, c_T$. For any vector $c_0$, we have:

$$\sum_{i=0}^{T} c_i(a_i^{\mathsf{BTPL}}) \le \sum_{i=0}^{T} c_i(a_T^{\mathsf{BTPL}}) \le \sum_{i=0}^{T} c_i(a_T^{\mathsf{BTL}}) \tag{5}$$

where the first inequality exhibits that BTPL has zero or negative regret on the cost sequence $c_0, \cdots, c_T$ and is proved similar to Lemma D.2. The second inequality holds since $a_T^{\mathsf{BTPL}}$ is the optimal action in hindsight given cost sequence $c_0, c_1, \cdots, c_T$. Therefore,

$$\sum_{i=1}^{T} c_i(a_i^{\mathsf{BTPL}}) \le \sum_{i=1}^{T} c_i(a_T^{\mathsf{BTL}}) + c_0(a_T^{\mathsf{BTL}}) - c_0(a_0^{\mathsf{BTPL}}) \tag{6}$$

The left-hand side of Equation (6) is the cost of BTPL. The first term on the right-hand side is the cost of the optimal action in hindsight. Term $c_0(a_T^{\mathsf{BTL}}) - c_0(a_0^{\mathsf{BTPL}}) \le \max_{a\in\mathcal{A}} X_a$. Therefore, the expected regret of BTPL is at most $\mathbb{E}[\max_{a\in\mathcal{A}} X_a]$ and the proof is complete. $\qquad\square$

The following lemma shows that the cost paid by FTPLB$^\star$ at time $t$ is close to the cost paid by BTPL at time $t$. Let $a_t^{\mathsf{FTPLB}^\star}$ denote the action taken by FTPLB$^\star$ at time $t$, then $a_t^{\mathsf{FTPLB}^\star} = \arg\min_{a \in \mathcal{A}} \sum_{i=0}^{B\lfloor (t-1)/B \rfloor} c_i(a) - X_a$.

**Lemma E.3.** *For any time $1 \le t \le T$, $\Pr[a_t^{\mathsf{FTPLB}^\star} \ne a_t^{\mathsf{BTPL}}] \le \varepsilon B$.*

*Proof.* We prove that for any $a^* \in \mathcal{A}$, $\Pr[a_t^{\mathsf{BTPL}} = a^* \mid a_t^{\mathsf{FTPLB}^\star} = a^*] \ge 1 - \varepsilon B$ and this will imply lemma. This is the case since

$$\Pr[a_t^{\mathsf{FTPLB}^\star} = a_t^{\mathsf{BTPL}}] = \sum_{a^* \in \mathcal{A}} \Pr[a_t^{\mathsf{BTPL}} = a^* \mid a_t^{\mathsf{FTPLB}^\star} = a^*] \Pr[a_t^{\mathsf{FTPLB}^\star} = a^*]$$

and the above would imply the RHS is at least $1 - \varepsilon B$.

First, we define some notations. For any $a \in \mathcal{A}$, let $U_a = \sum_{i=1}^{B\lfloor (t-1)/B \rfloor} c_i(a)$, and $v_a = \sum_{i=B\lfloor (t-1)/B \rfloor + 1}^{t} c_i(a)$. Now, $a_t^{\mathsf{FTPLB}^\star} = a^*$ implies that:

$$U_{a^*} - X_{a^*} \le U_a - X_a$$

Therefore for all $a \in \mathcal{A}$:

$$X_{a^*} \ge U_{a^*} - U_a + X_a$$

Similarly, we have $a_t^{\mathsf{BTPL}} = a^*$ if for all $a \in \mathcal{A}$:

$$X_{a^*} \ge U_{a^*} - U_a + X_a + (v_{a^*} - v_a)$$

Next, condition on $X_a = x_a$ for all $a \ne a^*$. Given these $x_a$ values, define $V = \max_{a \ne a^*} x_a - U_a$ and $v = \max_{a \ne a^*} v_{a^*} - v_a$, since all the $c_t(.)$ values are at most 1, $v \le B$. Then it is the case that:

$$\Pr_{X_{a^*}}[a_t^{\mathsf{BTPL}} = a^* \mid a_t^{\mathsf{FTPLB}^\star} = a^*, X_a = x_a, a \ne a^*] \ge \Pr_{X_{a^*}}[X_{a^*} \ge V + v + U_{a^*} \mid X_{a^*} \ge V + U_{a^*}]$$

where the probability is only over the randomness in $X_{a^*}$ since that is the only random variable remaining. By the memorylessness property of geometric random variables, we show the RHS is $(1 - \varepsilon)^B$. First, we remind the reader of the memorylessness property of random variables:

**Fact E.4** (Memorylessness of Geometric Random Variables). *Let $X \sim \mathbf{Geo}(p)$, then for any parameter $k$, we have:*

$$\Pr[X \ge k + 1 \mid X \ge k] = \frac{\Pr[X \ge k + 1]}{\Pr[X \ge k]} = (1 - p)$$

*This is the case since $X \sim \mathbf{Geo}(p)$ implies $\Pr[X \ge t] = (1 - p)^{t-1}$ as the first $t - 1$ experiments must fail.*

Therefore, we get that for any conditioning of $X_a = x_a$, for $a \ne a^*$, we have:

$$\Pr_{X_{a^*}}[a_t^{\mathsf{BTPL}} = a^* \mid a_t^{\mathsf{FTPLB}^\star} = a^*, X_a = x_a, a \ne a^*] \ge (1 - \varepsilon)^B \ge (1 - \varepsilon B)$$

which implies:

$$\Pr[a_t^{\mathsf{BTPL}} = a^* \mid a_t^{\mathsf{FTPLB}^\star} = a^*] \ge 1 - \varepsilon B$$

this yields the lemma. $\qquad\square$

Now we are ready to prove Theorem E.1.

*Proof of Theorem E.1.* First, we argue that $\mathbb{E}[\mathsf{Cost}(\mathsf{FTPLB}^\star)] \le \mathbb{E}[\mathsf{Cost}(\mathsf{BTPL})] + \varepsilon BT$.

$$\mathsf{Cost}(\mathsf{FTPLB}^\star) - \mathsf{Cost}(\mathsf{BTPL}) = \sum_{t=1}^{T} c_t(a_t^{\mathsf{FTPLB}^\star}) - c_t(a_t^{\mathsf{BTPL}})$$

For any $1 \le t \le T$, we see that $c_t(a_t^{\mathsf{FTPLB}^\star}) - c_t(a_t^{\mathsf{BTPL}}) \le 1$, and is indeed 0 if $a_t^{\mathsf{FTPLB}^\star} = a_t^{\mathsf{BTPL}}$. Therefore:

$$\mathbb{E}[\mathsf{Cost}(\mathsf{FTPLB}^\star)] - \mathbb{E}[\mathsf{Cost}(\mathsf{BTPL})] \le \sum_{t=1}^{T} \Pr[a_t^{\mathsf{FTPLB}^\star} \ne a_t^{\mathsf{BTPL}}] \le \varepsilon BT$$

where the last inequality holds by Lemma E.3.

By Lemma E.2, we know that for any expert $a \in \mathcal{A}$:

$$\mathbb{E}[\mathsf{Cost}(\mathsf{BTPL})] \leq \mathsf{Cost}(a) + \mathbb{E}[\max_{a \in \mathcal{A}} X_a]$$

We can bound the expectation of the maximum of geometric random variables using the following fact:

**Fact E.5** (Expectation of maximum of geometric random variables). *Let $X_1, \cdots, X_n \sim \mathbf{Geo}(p)$ be iid geometric random variables. Then, $\mathbb{E}[\max_i X_i] \leq 1 + \frac{H_n}{p}$, where $H_n$ is the $n^{th}$ Harmonic number.*

which implies that for any expert $a \in \mathcal{A}$:

$$\mathbb{E}[\mathsf{Cost}(\mathsf{BTPL})] \leq \mathsf{Cost}(a) + \frac{\ln n}{\varepsilon}$$

Finally, for any expert $a \in \mathcal{A}$:

$$\mathbb{E}[\mathsf{Cost}(\mathsf{FTPLB}^\star)] \leq \mathsf{Cost}(a) + \varepsilon BT + \frac{\ln n}{\varepsilon}$$

$\square$

Now we are ready to prove Theorem 4.1.

**Theorem 4.1** (Regret and Replicability Guarantees of FTPLB$^\star$). *FTPLB$^\star(\varepsilon, B)$ is adversarially $\rho$-replicable and achieves regret $\widetilde{\mathcal{O}}(T^{5/6} \ln^{5/6}(n) \rho^{-1/3})$ when $B = \left( \frac{8\sqrt{2(\frac{\log(8T/\rho)}{\log n} + 1)} \ln(n)T}{\rho} \right)^{2/3}$ and $\varepsilon = \sqrt{\frac{\ln n}{BT}}$.*

*Proof.* First, we bound the probability of non-replicability. Consider two different trajectories $c_1, \cdots, c_t$ and $c_1', \cdots, c_t'$ where each $c_i, c_i' \sim \mathcal{D}_i$. For each $a \in \mathcal{A}$, let function $f_a : \mathcal{C}_1 \times \cdots \times \mathcal{C}_t \to \mathbb{R}$ where $f_a(c_1, \cdots, c_t) = \sum_{i=1}^t c_i(a)$. Then for each $a \in \mathcal{A}$, $f_a$ satisfies the bounded difference property. For all $i \in [t]$ and all $c_1 \in \mathcal{C}_1, c_2 \in \mathcal{C}_2, \cdots, c_t \in \mathcal{C}_t$, since we assume the $\ell_\infty$ of all cost vectors is at most 1(this is the case since we assume cost of each expert at each time $t$ is in $[0, 1]$), we have:

$$\sup_{c_i' \in \mathcal{C}_i} |f_a(c_1, \cdots, c_{i-1}, c_i, c_{i+1}, \cdots, c_t) - f_a(c_1, \cdots, c_{i-1}, c_i', c_{i+1}, \cdots, c_t)| \leq 2$$

Hence by McDiarmid's inequality (see Theorem B.2),

$$\Pr\left[ \left| f_a(c_1, \ldots, c_t) - E[f_a(c_1, \ldots, c_t)] \right| > \varepsilon \right] \leq 2\exp\left( \frac{-2\varepsilon^2}{4t} \right).$$

Therefore, for each expert $a \in \mathcal{A}$:

$$\Pr\left( \left| f_a(c_{1:t}) - \mathbb{E}[f_a(c_{1:t})] \right| \geq p\sqrt{T \ln n} \right) \leq \Pr\left( \left| f_a(c_{1:t}) - \mathbb{E}[f_a(c_{1:t})] \right| \geq p\sqrt{t \ln n} \right)$$

$$\leq 2\exp\left( \frac{-p^2 \ln n}{2} \right) = \frac{2}{n^{p^2/2}}$$

By triangle's inequality and union bound:

$$\Pr\left( \left| f_a(c_{1:t}) - f_a(c_{1:t}') \right| \geq 2p\sqrt{T \ln n} \right) \leq \frac{4}{n^{p^2/2}} \tag{7}$$

By taking the union bound over all experts in $\mathcal{A}$:

$$\Pr\left(\exists a \in \mathcal{A} : \left|f_a(c_{1:t}) - f_a(c'_{1:t})\right| \geq 2p\sqrt{T \ln n}\right) \leq \frac{4n}{n^{p^2/2}} = \frac{\rho}{2T} \tag{8}$$

where the last inequality holds when $p = \sqrt{2(\frac{\log(8T/\rho)}{\log n} + 1)}$.

At a step $t$, where $t$ is a transition point, i.e. $t = \{B+1, 2B+1, \cdots, \}$, let $a_t^{\mathsf{FTPLB}^\star}$ denote the action taken by $\mathsf{FTPLB}^\star$ given cost sequence $c_1, \cdots, c_{t-1}$, and $b_t^{\mathsf{FTPLB}^\star}$ denote the action taken by $\mathsf{FTPLB}^\star$ given cost sequence $c'_1, \cdots, c'_{t-1}$. We prove that for any $a^* \in \mathcal{A}$,

$$\Pr[b_t^{\mathsf{FTPLB}^\star} = a^* \mid a_t^{\mathsf{FTPLB}^\star} = a^*] \geq 1 - 4\varepsilon p\sqrt{T \ln n}$$

This would imply:

$$\Pr[a_t^{\mathsf{FTPLB}^\star} = b_t^{\mathsf{FTPLB}^\star}] = \sum_{a^* \in \mathcal{A}} \Pr[b_t^{\mathsf{FTPLB}^\star} = a^* \mid a_t^{\mathsf{FTPLB}^\star} = a^*]\Pr[a_t^{\mathsf{FTPLB}^\star} = a^*] \geq 1 - 4\varepsilon p\sqrt{T \ln n}$$

Now, $a_t^{\mathsf{FTPLB}^\star} = a^*$ implies that:

$$f_{a^*}(c_{1:t-1}) - X_{a^*} \leq f_a(c_{1:t-1}) - X_a$$

Therefore for all $a \in \mathcal{A}$:

$$X_{a^*} \geq f_{a^*}(c_{1:t-1}) - f_a(c_{1:t-1}) + X_a$$

Now if $X_{a^*} \geq f_{a^*}(c_{1:t-1}) - f_a(c_{1:t-1}) + X_a + 4p\sqrt{T \ln n}$ and for all experts $a \in \mathcal{A}$, $|f_a(c_{1:t-1}) - f_a(c'_{1:t-1})| \leq 2p\sqrt{T \ln n}$, then it is the case that:

$$X_{a^*} \geq f_{a^*}(c'_{1:t-1}) - f_a(c'_{1:t-1}) + X_a$$

and therefore $b_t^{\mathsf{FTPLB}^\star} = a^*$.

Now, we get that for any conditioning of $X_a = x_a$, for $a \neq a^*$, by the memorylessness property of geometric random variables we have:

$$\Pr_{X_{a^*}}[b_t^{\mathsf{FTPLB}^\star} = a^* \mid a_t^{\mathsf{FTPLB}^\star} = a^*, X_a = x_a, a \neq a^*] \geq (1-\varepsilon)^{4p\sqrt{T \ln n}} \geq (1 - 4\varepsilon p\sqrt{T \ln n})$$

which implies:

$$\Pr[b_t^{\mathsf{FTPLB}^\star} = a^* \mid a_t^{\mathsf{FTPLB}^\star} = a^*] \geq 1 - 4\varepsilon p\sqrt{T \ln n}$$

This implies that the probability of non-replicability at a transition point $t$, is at most $\frac{\rho}{2T} + 4\varepsilon p\sqrt{T \ln n}$. By taking union bound over all the transition points $t = \{B+1, 2B+1, \cdots\}$ the total probability of non-replicability is at most $(\frac{\rho}{2T} + 4\varepsilon p\sqrt{T \ln n})\frac{T}{B}$. Consequently, we can set values of $B, \varepsilon$ such that the probability of non-replicability over all the time-steps is at most $\rho$:

$$(\frac{\rho}{2T} + 4\varepsilon p\sqrt{T \ln n})\frac{T}{B} \leq \rho$$

in order to achieve this, it suffices to have:

$$\frac{4\varepsilon p\sqrt{\ln n}T^{3/2}}{B} = \rho/2$$

or equivalently:

$$\varepsilon = \frac{B\rho}{8p\sqrt{\ln n}T^{3/2}}$$

Additionally, in order to minimize regret by Theorem E.1, we need to set $\varepsilon = \sqrt{\frac{\ln n}{BT}}$. Now by setting

$$\frac{B\rho}{8p\sqrt{\ln n}T^{3/2}} = \sqrt{\frac{\ln n}{BT}}$$

it implies that

$$B = \left( \frac{8\sqrt{2(\frac{\log(8T/\rho)}{\log n} + 1)\ln(n)T}}{\rho} \right)^{2/3} = \widetilde{O}(T^{2/3}\ln^{2/3}(n)(1/\rho)^{2/3})$$

Finally, plugging in the value of $B$ in Theorem E.1, gives a regret bound of $\mathcal{O}(\sqrt{BT\ln n}) = \widetilde{\mathcal{O}}(T^{5/6}\ln^{5/6}(n)\rho^{-1/3})$. $\qquad\qquad\qquad\qquad\qquad\qquad\qquad\qquad\qquad\qquad\qquad\qquad\qquad\square$

## F  Missing Proofs of Section 5

---
**Algorithm 7** General framework for converting a learning algorithm $\mathsf{ALG}_{int}$ to an adversarially $\rho$-replicable algorithm $\mathsf{ALG}_{ext}$ when $\mathsf{ALG}_{int}$ gets a bounded $\ell_1$ cost vector as input

---
**Input:** Sequence $S = \{c_1, \cdots, c_T\}$ arriving one by one over the time. $\varepsilon > 0$ and block size $B$ and a learning algorithm $\mathsf{ALG}$.

34  Let $a_1$ be a random action picked from $\mathcal{A}$.
35  **for** $t : 1, \cdots, T$ **do**
36      **if** $t - 1$ *is a multiple of $B$ and $t > 1$* **then**
        // the $u^{th}$ time block starts with $u = (t-1)/B + 1$.
37          Sample $p_t, p'_t \sim [0, \frac{1}{\varepsilon})^n$ uniformly at random.
38          Let $G \leftarrow \{p_t + \frac{1}{\varepsilon}z \mid z \in \mathcal{Z}^n\}$.
39          Let $G' \leftarrow \{p'_t + \frac{1}{\varepsilon}z \mid z \in \mathcal{Z}^n\}$.
40          $g_{t-1} \leftarrow G \cap \left( c_{1:t-1} + [-\frac{1}{2\varepsilon}, \frac{1}{2\varepsilon})^n \right)$.
41          $g'_{t-1-B} \leftarrow G' \cap \left( c_{1:t-1-B} + [-\frac{1}{2\varepsilon}, \frac{1}{2\varepsilon})^n \right)$.
42          $a_t \leftarrow \mathsf{ALG}_{int}\left( \frac{1}{B + \frac{2n}{\varepsilon}}(g_{t-1} - g'_{t-1-B}) \right)$.
43      **else**
        // Stay with the previous action.
44          $a_t \leftarrow a_{t-1}$

---

In order to prove Theorem 5.1, first we prove the following lemma holds which bounds the regret of Algorithm 7.

**Lemma F.1.** *Assume that that the cost function $\mathsf{Cost}(a, c)$ is linear in $c$, i.e., $\mathsf{Cost}(a, c_1 + c_2) = \mathsf{Cost}(a, c_1) + \mathsf{Cost}(a, c_2)$. In Algorithm 7, the regret of $\mathsf{ALG}_{ext}$ over $T$ timesteps is at most:*

$$\mathbb{E}[\mathsf{Reg}_T(\mathsf{ALG}_{ext})] \leq (B + 2n/\varepsilon)\mathsf{Reg}_{T/B}(\mathsf{ALG}_{int})$$

*Proof.* First, we define the following sequences: $S = \{c_1, \cdots, c_T\}$, $\hat{S} = \{g_B, (g_{2B} - g'_B), \cdots, (g_{B\lfloor T/B \rfloor} - g'_{B(\lfloor T/B \rfloor - 1)})\}$, $\tilde{S} = \hat{S}/(B + 2n/\varepsilon)$. $\tilde{S}$ is the sequence that we feed into $\mathsf{ALG}_{int}$. Let $\mathcal{R}_{int}$ denote the internal randomness of $\mathsf{ALG}_{int}$ and $\mathcal{R}_{ext}$ be the randomness of $\mathsf{ALG}_{ext}$ over random offsets $p_t, p'_t$ at each transition point $t$. Let $\mathsf{Cost}(\mathsf{ALG}, S)$ define the cost of running $\mathsf{ALG}$ on sequence $S$ which is equal to $\mathsf{Cost}(\mathsf{ALG}, S) = \sum_i cost(a_i^{\mathsf{ALG}}, S_i)$, where $a_i^{\mathsf{ALG}}$ is the $i^{th}$ action taken by $\mathsf{ALG}$, and $cost(a_i^{\mathsf{ALG}}, S_i)$ is the incurred cost when action $a_i^{\mathsf{ALG}}$ is chosen and the cost vector is $S_i$.

By the regret guarantees of $\mathsf{ALG}_{int}$ over $T/B$ timesteps, it is the case that for any fixed sequence $X$ and action $a \in \mathcal{A}$:

$$\mathbb{E}_{\mathcal{R}_{int}}[\mathsf{Cost}(\mathsf{ALG}_{int}, X)] \leq \mathsf{Cost}(a, X) + \mathsf{Reg}_{T/B}(\mathsf{ALG}_{int}) \qquad (9)$$

and therefore:

$$\mathbb{E}_{\mathcal{R}_{int}, \mathcal{R}_{ext}}[\mathsf{Cost}(\mathsf{ALG}_{int}, \tilde{S})] \leq \mathbb{E}_{\mathcal{R}_{ext}}[\mathsf{Cost}(a, \tilde{S})] + \mathsf{Reg}_{T/B}(\mathsf{ALG}_{int}) \qquad (10)$$

Note that $\tilde{S}$ is a probabilistic sequence that depends on $\mathcal{R}_{ext}$. Moreover, $\mathsf{Cost}(\mathsf{ALG}_{int}, \tilde{S})$ is a random variable that depends on the randomness of $\mathcal{R}_{int}$ and the random sequence $\tilde{S}$. Since $\hat{S}$ is a scaled version of $\tilde{S}$, then:

$$\underset{\mathcal{R}_{int}, \mathcal{R}_{ext}}{\mathbb{E}} [\mathsf{Cost}(\mathsf{ALG}_{int}, \hat{S})] = (B + 2n/\varepsilon) \underset{\mathcal{R}_{int}, \mathcal{R}_{ext}}{\mathbb{E}} [\mathsf{Cost}(\mathsf{ALG}_{int}, \tilde{S})] \tag{11}$$

Next, we bound the total cost of $\mathsf{ALG}_{ext}$ on the true cost sequence $S$. WLOG we assume $T$ is a multiple of $B$, if not, we add zero cost vectors to $S$ to make $T$ a multiple of $B$. Then:

$$\underset{\mathcal{R}_{int}, \mathcal{R}_{ext}}{\mathbb{E}} [\mathsf{Cost}(\mathsf{ALG}_{ext}, S)] = \sum_{u=1}^{T/B} \underset{\mathcal{R}_{int}, \mathcal{R}_{ext}}{\mathbb{E}} [\mathsf{Cost}(a_u, c_{(u-1)B:uB})] \tag{12}$$

where in Equation (12), $a_u$ is the action taken by $\mathsf{ALG}_{ext}$ in the $u$-th block, and $c_{(u-1)B:uB}$ is the cumulative cost vector within the time-interval $((u-1)B, uB]$. $\mathsf{Cost}(\mathsf{ALG}_{ext}, S)$ is a random variable that depends on $\mathcal{R}_{int}$ and $\mathcal{R}_{ext}$. This is the case since the action $a_u$ that is chosen by $\mathsf{ALG}_{int}$ depends on $\mathcal{R}_{int}$ and the probabilistic rounded cost vector that is fed into $\mathsf{ALG}_{int}$ which depends on $\mathcal{R}_{ext}$.

Furthermore:

$$\underset{\mathcal{R}_{int}, \mathcal{R}_{ext}}{\mathbb{E}} [\mathsf{Cost}(\mathsf{ALG}_{int}, \hat{S})] = \sum_{u=1}^{T/B} \underset{\mathcal{R}_{int}, \mathcal{R}_{ext}}{\mathbb{E}} [\mathsf{Cost}(a_u, g_{uB} - g'_{(u-1)B})] \tag{13}$$

$$= \sum_{u=1}^{T/B} \underset{\mathcal{R}_{int}}{\mathbb{E}} \left[ \underset{\mathcal{R}_{ext}}{\mathbb{E}} [\mathsf{Cost}(a_u, g_{uB} - g'_{(u-1)B})|a_u]] \right] = \sum_{u=1}^{T/B} \underset{\mathcal{R}_{int}}{\mathbb{E}} [\mathsf{Cost}(a_u, \underset{\mathcal{R}_{ext}}{\mathbb{E}} [g_{uB} - g'_{(u-1)B}|a_u])], \tag{14}$$

$$= \sum_{u=1}^{T/B} \underset{\mathcal{R}_{int}}{\mathbb{E}} [\mathsf{Cost}(a_u, c_{uB} - c_{(u-1)B})] = \sum_{u=1}^{T/B} \underset{\mathcal{R}_{int}}{\mathbb{E}} [\mathsf{Cost}(a_u, c_{(u-1)B:uB})]. \tag{15}$$

where Equation (14) holds by using the linearity of the cost function, and Equation (15) holds using the fact that $a_u$ is independent of both $g'_{(u-1)B}$ and $g_{uB}$. This is the case since $u = (t-1)/B + 1$ and at a transition point $t$, $a_t$ depends on $g_{t-1}$ and $g'_{t-1-B}$ implying that $a_u$ depends on $g_{(u-1)B}$ and $g'_{(u-2)B}$ (and not $g'_{(u-1)B}$ and $g_{uB}$). Putting together Equations (12) and (15) implies that:

$$\underset{\mathcal{R}_{int}, \mathcal{R}_{ext}}{\mathbb{E}} [\mathsf{Cost}(\mathsf{ALG}_{int}, \hat{S})] = \underset{\mathcal{R}_{int}, \mathcal{R}_{ext}}{\mathbb{E}} [\mathsf{Cost}(\mathsf{ALG}_{ext}, S)] \tag{16}$$

Now, combining Equations (11) and (16) implies:

$$\underset{\mathcal{R}_{int}, \mathcal{R}_{ext}}{\mathbb{E}} [\mathsf{Cost}(\mathsf{ALG}_{ext}, S)] = (B + 2n/\varepsilon) \underset{\mathcal{R}_{int}, \mathcal{R}_{ext}}{\mathbb{E}} [\mathsf{Cost}(\mathsf{ALG}_{int}, \tilde{S})] \tag{17}$$

Now, for any action $a \in \mathcal{A}$, we bound $\mathbb{E}_{\mathcal{R}_{ext}}[\mathsf{Cost}(a, \tilde{S})]$ as follows:

$$\underset{\mathcal{R}_{ext}}{\mathbb{E}} [\mathsf{Cost}(a, \tilde{S})] = (1/(B + 2n/\varepsilon)) \underset{\mathcal{R}_{ext}}{\mathbb{E}} [\mathsf{Cost}(a, \hat{S})] \tag{18}$$

$$= (1/(B + 2n/\varepsilon)) \sum_{u=1}^{T/B} \underset{\mathcal{R}_{ext}}{\mathbb{E}} [\mathsf{Cost}(a, g_{uB} - g'_{(u-1)B})] \tag{19}$$

$$= (1/(B + 2n/\varepsilon)) \sum_{u=1}^{T/B} \mathsf{Cost}(a, c_{(u-1)B:uB}) \tag{20}$$

$$= (1/(B + 2n/\varepsilon)) \mathsf{Cost}(a, c_{1:T}) \tag{21}$$

$$= (1/(B + 2n/\varepsilon)) \mathsf{Cost}(a, S) \tag{22}$$

Now, combining Equations (10), (17) and (22) implies that for any action $a \in \mathcal{A}$:

$$\frac{\mathbb{E}_{\mathcal{R}_{int}, \mathcal{R}_{ext}}[\mathsf{Cost}(\mathsf{ALG}_{ext}, S)]}{B + 2n/\varepsilon} \leq \frac{\mathsf{Cost}(a, S)}{B + 2n/\varepsilon} + \mathsf{Reg}_{T/B}(\mathsf{ALG}_{int})$$

which implies that for any sequence $S$ and any action $a \in \mathcal{A}$:

$$\mathbb{E}_{\mathcal{R}_{int}, \mathcal{R}_{ext}}[\mathsf{Cost}(\mathsf{ALG}_{ext}, S)] \leq \mathsf{Cost}(a, S) + (B + 2n/\varepsilon)\mathsf{Reg}_{T/B}(\mathsf{ALG}_{int})$$

and the proof is complete. □

**Theorem 5.1** (Regret and Replicability Guarantees of Algorithm 7). *Let $\rho > 0$ be a parameter, and assume that the cost function $\mathsf{Cost}(a, c)$ is linear in $c$, i.e., $\mathsf{Cost}(a, c_1 + c_2) = \mathsf{Cost}(a, c_1) + \mathsf{Cost}(a, c_2)$.*

*Suppose for two different input trajectories $c_1, \cdots, c_T$ and $d_1, \cdots, d_T$ where for each time-step $t$, $c_t, d_t \sim \mathcal{D}_i$, for each time step $t$, the $\ell_1$ distance between $c_{1:t}$ and $d_{1:t}$ is at most $m$ with probability at least $1 - \gamma$. Then, Algorithm 7 is adversarially $\rho$-replicable and achieves cumulative regret $\mathcal{O}\left(B\mathsf{Reg}_{T/B}(\mathsf{ALG}_{int})\right)$ when $\gamma$ is at most $\frac{\rho B}{4T}$, $B = \sqrt{\frac{8nmT}{\rho}}$ and $\varepsilon = 2n/B$.*

*Proof of Theorem 5.1.* First, we argue about the replicability. Here we assume that given two different cost sequences $c_1, \cdots, c_T$ and $d_1, \cdots, d_T$, where for each timestep $t$, $c_t, d_t$ are drawn from the same distribution, for each time step $t$, the $\ell_1$ distance between $c_{1:t}$ and $d_{1:t}$ is at most $m$ with probability at least $1 - \gamma$.

Let $g, g'$ denote the sequence of grid points selected given the cost sequence $\{c_1, \cdots, c_T\}$ and $q, q'$ denote the sequence of grid points selected given the cost sequence $\{d_1, \cdots, d_T\}$. First, we bound the probability of the event that $g_{t-1} \neq q_{t-1}$. This event happens iff the grid point in $c_{1:t-1} + [0, \frac{1}{\varepsilon}]^n$ is not in $d_{1:t-1} + [0, \frac{1}{\varepsilon}]^n$. We argue that for any $v \in \mathbb{R}^n$, the cubes $[0, 1/\varepsilon]^n$ and $v + [0, 1/\varepsilon]^n$ overlap in at least a $(1 - \varepsilon\ell_1(v))$ fraction. Take a random point $x \in [0, 1/\varepsilon]^n$. If $x \notin v + [0, 1/\varepsilon]^n$, then for some $i$, $x_i \notin v_i + [0, 1/\varepsilon]$ which happens with probability at most $\varepsilon\ell_1(v_i)$ for any particular $i$. By the union bound, we are done.

This implies the probability of the event that $g_{t-1} \neq q_{t-1}$ at a transition point $t$, i.e. $t - 1$ is a multiple of $B$, is at most $\gamma + \varepsilon m$. Similarly, the probability that $g'_{t-1-B} \neq q'_{t-1-B}$ is at most $\gamma + \varepsilon m$. By taking union bound over all the transition points $t = \{B + 1, 2B + 1, \cdots\}$ the total probability of non-replicability is at most $2(\gamma + \varepsilon m)\frac{T}{B}$. Consequently, we can set values of $B, \varepsilon$ such that the probability of non-replicability over all the time-steps is at most $\rho$:

$$\rho = 2(\gamma + \varepsilon m)\frac{T}{B}$$

Since $\gamma \leq \frac{\rho B}{4T}$, we need to set $\frac{2\varepsilon mT}{B} = \rho/2$. Now, we consider the regret. By Lemma F.1, the regret of Algorithm 7 is $(B + 2n/\varepsilon)\mathsf{Reg}_{T/B}(\mathsf{ALG}_{int})$, therefore, in order to minimize the regret, we need to set $B = 2n/\varepsilon$, which implies that $\rho = \frac{4\varepsilon mT}{B} = \frac{8nmT}{B^2}$. Consequently, $B = \sqrt{\frac{8nmT}{\rho}}$, and the regret of Algorithm 7 is $2B\mathsf{Reg}_{T/B}(\mathsf{ALG}_{int})$. □

## F.1 Proof of Corollary 5.2

*Proof.* By Lemma C.1, for two different trajectories $c_1, \cdots, c_t$ and $d_1, \cdots, d_t$ of $n$-dimensional vectors where for each $i$, $c_i, d_i \sim \mathcal{D}_i$ we can bound the $\ell_1$ distance of the cumulative costs of these two trajectories as follows:

$$\Pr\left[\ell_1(c_{1:t} - d_{1:t}) > p\sqrt{nT}\right] \leq \Pr\left[\ell_1(c_{1:t} - d_{1:t}) > p\sqrt{nt}\right] \leqslant \exp\left(-\frac{(p-2)^2 n}{2}\right) \quad (23)$$

where the right hand side is at most $\frac{\rho}{4T}$ when $p \geq \sqrt{\frac{2\log 4T/\rho}{n}} + 2$. Now, we can apply Theorem 5.1, where $m = \left(\sqrt{\frac{2\log 4T/\rho}{n}} + 2\right)\sqrt{nT}$ by Equation (23). In Algorithm 7, we use FTPL algorithm as the internal algorithm for $T/B$ timesteps that gives regret bound $\mathsf{Reg}_{\mathsf{ALG}_{int}} = \mathcal{O}(D\sqrt{T/B})$. By Theorem 5.1, the regret of Algorithm 7 is $\Omega(B\mathsf{Reg}_{T/B}(\mathsf{ALG}_{int}))$ when $B$ is set as $\sqrt{\frac{8mnT}{\rho}}$. Plugging in the values of $m$, $B$ and $\mathsf{Reg}_{\mathsf{ALG}_{int}}$ gives a total cumulative regret of $\widetilde{\mathcal{O}}(DT^{7/8}n^{3/8}\rho^{-1/4})$. □

# G   General framework when the internal algorithm takes as input bounded $\ell_\infty$ cost vectors

In this section, we consider a setting where the internal algorithm $\mathsf{ALG}_{int}$ takes as input a sequence of bounded $\ell_\infty$ cost vectors. We provide a general framework in Algorithm 8 that converts the internal algorithm to an adversarially $\rho$-replicable algorithm $\mathsf{ALG}_{ext}$ with bounded regret. The framework is similar to Algorithm 7, but with a different normalizing step. Similar to Algorithm 7, if $t-1$ is a multiple of the block size $B$, it picks a $n$-dimensional grid with spacing $1/\varepsilon$ where $\varepsilon$ is given as input, and a random offset $p_t \sim [0, 1/\varepsilon)^n$, where $n$ is the dimension of the space. Then it considers the cube $c_{1:t-1} + [0, 1/\varepsilon)^n$, where $c_{1:k} = \sum_{i=1}^{k} c_i$, and picks the unique grid points $g_{t-1}$ inside this cube. Furthermore, for a fresh random offset $p_t' \sim [0, 1/\varepsilon)^n$, it picks a new $n$-dimensional grid with spacing $1/\varepsilon$ and offset $p_t'$. Then, it considers the cube $c_{1:t-1-B} + [0, 1/\varepsilon)^n$ and picks the unique grid point $g_{t-1-B}'$ inside this cube. Then, it gives $\frac{1}{B+2/\varepsilon}(g_{t-1} - g_{t-1-B}')$ as the input argument to the algorithm $\mathsf{ALG}_{int}$. The normalizing factor makes sure that the input given to $\mathsf{ALG}_{int}$ has $\ell_\infty$ cost at most 1. Then $\mathsf{ALG}_{ext}$ plays the action returned by $\mathsf{ALG}_{int}$ for the next $B$ timesteps. We derive regret and replicability guarantees for Algorithm 8 in Theorem G.2. In order to prove Theorem G.2, first we bound the regret of Algorithm 8 in Lemma G.1. The steps of proving this lemma are similar to Lemma F.1, with taking into account the different normalizing factor.

---

**Algorithm 8** General framework for converting a learning algorithm $\mathsf{ALG}_{int}$ to an adversarially $\rho$-replicable algorithm $\mathsf{ALG}_{ext}$ when $\mathsf{ALG}_{int}$ gets a bounded $\ell_\infty$ cost vector as input

---

**Input:** Sequence $S = \{c_1, \cdots, c_T\}$ arriving one by one over the time. $\varepsilon > 0$ and block size $B$ and regret minimization algorithm $\mathsf{ALG}$.

45  Let $a_1$ be a random expert picked from $\mathcal{A}$.
46  **for** $t : 1, \cdots, T$ **do**
47   **if** $t-1$ *is a multiple of* $B$ *and* $t > 1$ **then**
48    Sample $p_t, p_t' \sim [0, \frac{1}{\varepsilon})^n$ uniformly at random.
49    Let $G \leftarrow \{p_t + \frac{1}{\varepsilon}z \mid z \in \mathcal{Z}^n\}$.
50    Let $G' \leftarrow \{p_t' + \frac{1}{\varepsilon}z \mid z \in \mathcal{Z}^n\}$.
51    $g_{t-1} \leftarrow G \cap \left(c_{1:t-1} + [-\frac{1}{2\varepsilon}, \frac{1}{2\varepsilon}]^n\right)$.
52    $g_{t-1-B}' \leftarrow G' \cap \left(c_{1:t-1-B} + [-\frac{1}{2\varepsilon}, \frac{1}{2\varepsilon}]^n\right)$.
53    $a_t \leftarrow \mathsf{ALG}_{int}\left(\frac{1}{B+\frac{2}{\varepsilon}}(g_{t-1} - g_{t-1-B}')\right)$.
54   **else**
      // Stay with the previous action.
55    $a_t \leftarrow a_{t-1}$

---

**Lemma G.1.** *Assume that that the cost function* $\mathsf{Cost}(a, c)$ *is linear in* $c$, *i.e.,* $\mathsf{Cost}(a, c_1 + c_2) = \mathsf{Cost}(a, c_1) + \mathsf{Cost}(a, c_2)$. *In Algorithm 8, the regret of* $\mathsf{ALG}_{ext}$ *over* $T$ *timesteps is at most:*

$$\mathbb{E}[\mathsf{Reg}_T(\mathsf{ALG}_{ext})] \leq (B + 2/\varepsilon)\mathsf{Reg}_{T/B}(\mathsf{ALG}_{int})$$

*Proof.* First, we define the following sequences: $S = \{c_1, \cdots, c_T\}$, $\hat{S} = \{g_B, (g_{2B} - g_B'), \cdots, (g_{B\lfloor T/B \rfloor} - g_{B(\lfloor T/B \rfloor - 1)}')\}$, $\tilde{S} = \hat{S}/(B + 2/\varepsilon)$. $\tilde{S}$ is the sequence that we feed into $\mathsf{ALG}_{int}$. Let $\mathcal{R}_{int}$ denote the internal randomness of $\mathsf{ALG}_{int}$ and $\mathcal{R}_{ext}$ be the randomness of $\mathsf{ALG}_{ext}$ over random offsets $p_t, p_t'$ at each transition point $t$. Let $\mathsf{Cost}(\mathsf{ALG}, S)$ define the cost of running $\mathsf{ALG}$ on sequence $S$ which is equal to $\mathsf{Cost}(\mathsf{ALG}, S) = \sum_i cost(a_i^{\mathsf{ALG}}, S_i)$, where $a_i^{\mathsf{ALG}}$ is the $i^{th}$ action taken by $\mathsf{ALG}$, and $cost(a_i^{\mathsf{ALG}}, S_i)$ is the incurred cost when action $a_i^{\mathsf{ALG}}$ is chosen and the cost vector is $S_i$.

By the regret guarantees of $\mathsf{ALG}_{int}$ over $T/B$ timesteps, it is the case that for any fixed sequence $X$ and action $a \in \mathcal{A}$:

$$\underset{\mathcal{R}_{int}}{\mathbb{E}}\left[\mathsf{Cost}(\mathsf{ALG}_{int}, X)\right] \leq \mathsf{Cost}(a, X) + \mathsf{Reg}_{T/B}(\mathsf{ALG}_{int}) \tag{24}$$

and therefore:

$$\underset{\mathcal{R}_{int}, \mathcal{R}_{ext}}{\mathbb{E}}\left[\mathsf{Cost}(\mathsf{ALG}_{int}, \tilde{S})\right] \leq \underset{\mathcal{R}_{ext}}{\mathbb{E}}\left[\mathsf{Cost}(a, \tilde{S})\right] + \mathsf{Reg}_{T/B}(\mathsf{ALG}_{int}) \tag{25}$$

Note that $\tilde{S}$ is a probabilistic sequence that depends on $\mathcal{R}_{ext}$. Moreover, $\text{Cost}(\text{ALG}_{int}, \tilde{S})$ is a random variable that depends on the randomness of $\mathcal{R}_{int}$ and the random sequence $\tilde{S}$. Since $\hat{S}$ is a scaled version of $\tilde{S}$, then:

$$\underset{\mathcal{R}_{int}, \mathcal{R}_{ext}}{\mathbb{E}} [\text{Cost}(\text{ALG}_{int}, \hat{S})] = (B + 2/\varepsilon) \underset{\mathcal{R}_{int}, \mathcal{R}_{ext}}{\mathbb{E}} [\text{Cost}(\text{ALG}_{int}, \tilde{S})] \tag{26}$$

Next, we bound the total cost of $\text{ALG}_{ext}$ on the true cost sequence $S$. WLOG we assume $T$ is a multiple of $B$, if not, we add zero cost vectors to $S$ to make $T$ a multiple of $B$. Then:

$$\underset{\mathcal{R}_{int}, \mathcal{R}_{ext}}{\mathbb{E}} [\text{Cost}(\text{ALG}_{ext}, S)] = \sum_{u=1}^{T/B} \underset{\mathcal{R}_{int}, \mathcal{R}_{ext}}{\mathbb{E}} [\text{Cost}(a_u, c_{(u-1)B:uB})] \tag{27}$$

where in Equation (27), $a_u$ is the action taken by $\text{ALG}_{ext}$ in the $u$-th block, and $c_{(u-1)B:uB}$ is the cumulative cost vector within the time-interval $((u-1)B, uB]$. $\text{Cost}(\text{ALG}_{ext}, S)$ is a random variable that depends on $\mathcal{R}_{int}$ and $\mathcal{R}_{ext}$. This is the case since the action $a_u$ that is chosen by $\text{ALG}_{int}$ depends on $\mathcal{R}_{int}$ and the probabilistic rounded cost vector that is fed into $\text{ALG}_{int}$ which depends on $\mathcal{R}_{ext}$.

Furthermore:

$$\underset{\mathcal{R}_{int}, \mathcal{R}_{ext}}{\mathbb{E}} [\text{Cost}(\text{ALG}_{int}, \hat{S})] = \sum_{u=1}^{T/B} \underset{\mathcal{R}_{int}, \mathcal{R}_{ext}}{\mathbb{E}} [\text{Cost}(a_u, g_{uB} - g'_{(u-1)B})] \tag{28}$$

$$= \sum_{u=1}^{T/B} \underset{\mathcal{R}_{int}, \mathcal{R}_{ext}}{\mathbb{E}} [\text{Cost}(a_u, g_{uB})] - \underset{\mathcal{R}_{int}, \mathcal{R}_{ext}}{\mathbb{E}} [\text{Cost}(a_u, g'_{(u-1)B})] \tag{29}$$

now using the fact that $a_u$ is independent of both $g'_{(u-1)B}$ and $g_{uB}$ we have (note that $a_u$ depends on $g_{(u-1)B}$ and $g'_{(u-2)B}$),

$$\tag{30}$$

$$= \sum_{u=1}^{T/B} \underset{\mathcal{R}_{int}, \mathcal{R}_{ext}}{\mathbb{E}} [\text{Cost}(a_u, c_{(u-1)B:uB})] \tag{31}$$

Putting together Equations (27) and (31) implies that:

$$\underset{\mathcal{R}_{int}, \mathcal{R}_{ext}}{\mathbb{E}} [\text{Cost}(\text{ALG}_{int}, \hat{S})] = \underset{\mathcal{R}_{int}, \mathcal{R}_{ext}}{\mathbb{E}} [\text{Cost}(\text{ALG}_{ext}, S)] \tag{32}$$

Now, combining Equations (26) and (32) implies:

$$\underset{\mathcal{R}_{int}, \mathcal{R}_{ext}}{\mathbb{E}} [\text{Cost}(\text{ALG}_{ext}, S)] = (B + 2/\varepsilon) \underset{\mathcal{R}_{int}, \mathcal{R}_{ext}}{\mathbb{E}} [\text{Cost}(\text{ALG}_{int}, \tilde{S})] \tag{33}$$

Now, for any action $a \in \mathcal{A}$, we bound $\mathbb{E}_{\mathcal{R}_{ext}}[\text{Cost}(a, \tilde{S})]$ as follows:

$$\underset{\mathcal{R}_{ext}}{\mathbb{E}} [\text{Cost}(a, \tilde{S})] = (1/(B + 2/\varepsilon)) \underset{\mathcal{R}_{ext}}{\mathbb{E}} [\text{Cost}(a, \hat{S})] \tag{34}$$

$$= (1/(B + 2/\varepsilon)) \sum_{u=1}^{T/B} \underset{\mathcal{R}_{ext}}{\mathbb{E}} [\text{Cost}(a, g_{uB} - g'_{(u-1)B})] \tag{35}$$

$$= (1/(B + 2/\varepsilon)) \sum_{u=1}^{T/B} \text{Cost}(a, c_{(u-1)B:uB}) \tag{36}$$

$$= (1/(B + 2/\varepsilon)) \text{Cost}(a, c_{1:T}) \tag{37}$$

$$= (1/(B + 2/\varepsilon)) \text{Cost}(a, S) \tag{38}$$

Now, combining Equations (25), (33) and (38) implies that for any action $a \in \mathcal{A}$:

$$\frac{\mathbb{E}_{\mathcal{R}_{int}, \mathcal{R}_{ext}}[\mathsf{Cost}(\mathsf{ALG}_{ext}, S)]}{B + 2/\varepsilon} \leq \frac{\mathsf{Cost}(a, S)}{B + 2/\varepsilon} + \mathsf{Reg}_{T/B}(\mathsf{ALG}_{int})$$

which implies that for any sequence $S$ and any action $a \in \mathcal{A}$:

$$\mathbb{E}_{\mathcal{R}_{int}, \mathcal{R}_{ext}}[\mathsf{Cost}(\mathsf{ALG}_{ext}, S)] \leq \mathsf{Cost}(a, S) + (B + 2/\varepsilon)\mathsf{Reg}_{T/B}(\mathsf{ALG}_{int})$$

and the proof is complete. □

**Theorem G.2.** *(Regret and Replicability Guarantees of Algorithm 8) Let $\rho > 0$ be a parameter, and assume that the cost function $\mathsf{Cost}(a, c)$ is linear in $c$, i.e., $\mathsf{Cost}(a, c_1 + c_2) = \mathsf{Cost}(a, c_1) + \mathsf{Cost}(a, c_2)$. Suppose for two different trajectories $c_1, \cdots, c_T$ and $d_1, \cdots, d_T$ where for each timestep $t$, $c_t, d_t \sim \mathcal{D}_i$, we can prove that for each time step $t$, the $\ell_1$ distance between $c_{1:t}$ and $d_{1:t}$ is at most $m$ with probability at least $1 - \gamma$. Then, Algorithm 8 is adversarially $\rho$-replicable and achieves cumulative regret $\Omega\left(B\mathsf{Reg}_{T/B}(\mathsf{ALG}_{int})\right)$ when $\gamma$ is at most $\frac{\rho B}{4T}$, $B = \sqrt{\frac{8mT}{\rho}}$ and $\varepsilon = 2/B$.*

*Proof of Theorem G.2.* First, we argue about the replicability. Here we assume that given two different cost sequences $c_1, \cdots, c_T$ and $d_1, \cdots, d_T$, where for each timestep $t$, $c_t, d_t$ are drawn from the same distribution, for each time step $t$, the $\ell_1$ distance between $c_{1:t}$ and $d_{1:t}$ is at most $m$ with probability at least $1 - \gamma$.

Let $g, g'$ denote the sequence of grid points selected given the cost sequence $\{c_1, \cdots, c_T\}$ and $q, q'$ denote the sequence of grid points selected given the cost sequence $\{d_1, \cdots, d_T\}$. First, we bound the probability of the event that $g_{t-1} \neq q_{t-1}$. This event happens iff the grid point in $c_{1:t-1} + [0, \frac{1}{\varepsilon}]^n$ is not in $d_{1:t-1} + [0, \frac{1}{\varepsilon}]^n$. We argue that for any $v \in \mathbb{R}^n$, the cubes $[0, 1/\varepsilon]^n$ and $v + [0, 1/\varepsilon]^n$ overlap in at least a $(1 - \varepsilon\ell_1(v))$ fraction. Take a random point $x \in [0, 1/\varepsilon]^n$. If $x \notin v + [0, 1/\varepsilon]^n$, then for some $i$, $x_i \notin v_i + [0, 1/\varepsilon]$ which happens with probability at most $\varepsilon\ell_1(v_i)$ for any particular $i$. By the union bound, we are done.

This implies the probability of the event that $g_{t-1} \neq q_{t-1}$ at a transition point $t$, i.e. $t - 1$ is a multiple of $B$, is at most $\gamma + \varepsilon m$. Similarly, the probability that $g'_{t-1-B} \neq q'_{t-1-B}$ is at most $\gamma + \varepsilon m$. By taking union bound over all the transition points $t = \{B + 1, 2B + 1, \cdots\}$ the total probability of non-replicability is at most $2(\gamma + \varepsilon m)\frac{T}{B}$. Consequently, we can set values of $B, \varepsilon$ such that the probability of non-replicability over all the time-steps is at most $\rho$:

$$\rho = 2(\gamma + \varepsilon m)\frac{T}{B}$$

Since $\gamma \leq \frac{\rho B}{4T}$, we need to set $\frac{2\varepsilon mT}{B} = \rho/2$. Now, we consider the regret. By Lemma G.1, the regret of Algorithm 8 is $(B + 2/\varepsilon)\mathsf{Reg}_{T/B}(\mathsf{ALG}_{int})$, therefore, in order to minimize the regret, we need to set $B = 2/\varepsilon$, which implies that $\rho = 2\varepsilon^2 mT = \frac{8mT}{B^2}$. Consequently, $B = \sqrt{\frac{8mT}{\rho}}$, and the regret of Algorithm 8 is $2B\mathsf{Reg}_{T/B}(\mathsf{ALG}_{int})$. □

## G.1 Application of Algorithm 8 to the experts problem

**Corollary G.3.** *In the experts problem setting, Algorithm 8 achieves adversarial $\rho$-replicability and cumulative regret $\widetilde{\mathcal{O}}(T^{7/8}n^{1/8}\rho^{-1/4})$.*

*Proof.* By Lemma C.1, for two different trajectories $c_1, \cdots, c_t$ and $d_1, \cdots, d_t$ of $n$-dimensional vectors where for each $i$, $c_i, d_i \sim \mathcal{D}_i$ we can bound the $\ell_1$ distance of the cumulative costs of these two trajectories as follows:

$$\Pr\left[\ell_1(c_{1:t} - d_{1:t}) > p\sqrt{nT}\right] \leq \Pr\left[\ell_1(c_{1:t} - d_{1:t}) > p\sqrt{nt}\right] \leqslant \exp\left(-\frac{(p-2)^2 n}{2}\right) \quad (39)$$

where the right hand side is at most $\frac{\rho}{4T}$ when $p \geq \sqrt{\frac{2\log 4T/\rho}{n}} + 2$. Now, we can apply Theorem G.2, where $m = \left(\sqrt{\frac{2\log 4T/\rho}{n}} + 2\right)\sqrt{nT}$ by Equation (39). In Algorithm 7, we use FTPL algorithm

as the internal algorithm for $T/B$ timesteps that gives regret bound $\mathsf{Reg}_{\mathsf{ALG}_{int}} = \mathcal{O}(\sqrt{\frac{T \log n}{B}})$. By Theorem G.2, the regret is $2B\mathsf{Reg}_{T/B}(\mathsf{ALG}_{int})$ when $B$ is set as $\sqrt{\frac{8mT}{\rho}}$. Plugging in the values of $m, B$ and $\mathsf{Reg}_{\mathsf{ALG}_{int}}$ gives a total cumulative regret of $\widetilde{\mathcal{O}}(T^{7/8}n^{1/8}\rho^{-1/4})$. □

## H iid-Replicability in the Experts Setting

In this setting we have $n$ experts, say $\mathcal{A} = \{1, 2 \ldots, n\}$. At each time step $t$ we get a cost profile: $c_t = (c_t(1), c_t(2), \ldots, c_t(n)) \in [0, 1]^n$ with $|c_t|_\infty \leq 1$. We want our algorithm to be iid $\rho$-replicable with worst case regret $K$. Recalling from Section 2, this entails:

1. (*Worst-Case Regret*) For any sequence of costs $S = (c_1, c_2, \ldots, c_T)$ the maximum expected regret (over internal randomness $R$) suffered by the online algorithm is at most $K$, ie,
$$\mathop{\mathbb{E}}_{R \leftarrow \mathcal{R}}[\mathsf{Reg}(S, R)] \leq K,$$
where $\mathsf{Reg}(S, R)$ denotes the regret of the algorithm on the sequence $S$ and using internal randomness $R$.

2. (*Replicability*) Let $S_1, S_2$ be two independent cost vectors of length $T$ sampled iid from $\mathcal{D}^{\otimes T}$, where $\mathcal{D}$ is an unknown cost distribution over $[0, 1]^n$. Further, let $R$ be the internal randomness of the algorithm. Then,
$$\mathop{\mathrm{Pr}}_{S_1, S_2, R}[\mathsf{ALG}(S_1, R) = \mathsf{ALG}(S_2, R)] \geq 1 - \rho.$$

The optimal situation would be to have an iid $\rho$-replicable algorithm with worst case regret $\mathcal{O}(\sqrt{T \ln n} \times f(1/\rho))$ where $f$ is a mildly growing function. It turns out that even with $n = 2$ we require $f(1/\rho) \gtrsim 1/\rho$. This is because we can embed the coin problem from Impagliazzo et al. [2022] into an instance of regret minimization with $n = 2$. The coin problem is as follows: given $T$ iid samples from a coin whose bias is promised to be either $1/2 + \tau$ or $1/2 - \tau$ (for some $\tau \in [0, 1/2]$), we need to identify which is the case with probability at least $1 - \delta$ for some $\delta < 1/16$, and while being replicable with probability at least $1 - \rho$. In such a case from Impagliazzo et al. [2022] we have $\rho \geq \Omega\left(\frac{1}{\tau\sqrt{T}}\right)$. We can embed the coin problem in the experts setting by letting $\tau \approx K/T$ ($K$ is the regret upper bound) (see Theorem I.2 for more details). Hence, we get that $K > \Omega\left(\frac{\sqrt{T}}{\rho}\right)$.

It turns out that we can match this bound up to logarithmic factors. Specifically, we design an iid $\rho$-replicable algorithm (Algorithm 9) with
$$K \leq \mathcal{O}\left(\frac{1}{\rho} \times (\log\log(T))^2 \times \log\left(\frac{\log\log(T)}{\rho}\right) \times \sqrt{T \ln(n)}\right).$$

Now, we turn to explaining the ideas behind Algorithm 9. In the discussion below, we ignore various logarithmic factors such as $\sqrt{\log n}$ and think of $\rho > 0$ as a small fixed constant.

**Performance under iid cost sequence:** Let us first focus on achieving low regret when we are promised that the cost sequence $S = (c_1, c_2, \ldots, c_T)$ is sampled from $\mathcal{D}^{\otimes T}$, for some unknown distribution $\mathcal{D}$ over $[0, 1]^n$. We can always use a standard regret minimizing algorithm for this purpose such as FTPL: however, such an algorithm will not satisfy the replicability properties. Hence, we need to adopt a different approach.

The idea is that if we observe the experts for the first $L^{(1)} = \sqrt{T}$ time steps, we have a good approximation of the expected cost of each expert $a \in \mathcal{A}$, ie, $\mu_a := \mathop{\mathbb{E}}_{c \leftarrow \mathcal{D}}[c(a)]$, to within an accuracy of $T^{-1/4}$, using the empirical average $\sum_{j=1}^{L^{(1)}} c_j(a)/L^{(1)}$. (The standard deviation of the empirical average is of the order $T^{-1/4}$.) During these times we always select a fixed expert, say $a^{(1)} = 1$. Further, we accrue a regret of at most $\sqrt{T}$ during this block of $L^{(1)}$ steps.

Now, using our estimates of the $\mu_a$'s we select expert $a^{(2)} = \arg\min_{a \in \mathcal{A}} \sum_{j=1}^{L^{(1)}} c_j(a)$ for the next $L^{(2)} = T^{3/4}$ time steps. During these $T^{3/4}$ time steps, compared to an expert $a \in \mathcal{A}$, we accrue a regret of at most roughly $T^{-1/4} \times T^{3/4} = T^{1/2}$: this is because $\mu_{a^{(2)}} - \mu_a \lesssim T^{-1/4}$.

Now, we re-update our estimates of $\mu_a$'s using the new empirical averages and our accuracy is now roughly $T^{-3/8}$. This allows us to select expert $a^{(3)} = \arg\min_{a \in \mathcal{A}} \sum_{j=1}^{L^{(2)}} c_j(a)$ for the next block of $T^{7/8}$ time steps while only accruing roughly $\sqrt{T}$ more in regret. Continuing in this fashion we end up using roughly $\log\log(T)$ blocks and hence the regret accrued at the end of $T$ steps is at most $O(\log\log(T)\sqrt{T})$.

*But how do we make the above algorithm replicable?* The above strategy is unfortunately not replicable. To see this consider the situation when $\mu_1 = \mu_2 = \ldots = \mu_n$. In this case $a^{(2)}$ is equally likely to be any of the $n$ experts. To overcome the above hurdle we add some noise to each expert at the end of each block. Specifically, after time $L^{(1)}$, when deciding expert $a^{(2)}$, we add **Geo**$(\varepsilon \approx T^{-1/4})$ noise to each expert independently. The added noise helps us ensure that in two different samples of $c_1, \ldots, c_{L^{(1)}}$ from $\mathcal{D}^{\otimes L^{(1)}}$, we choose the same $a^{(2)}$ with high probability: we expect that with high probability $\forall a \in \mathcal{A} : \sum_{j=1}^{L^{(1)}} c_j(a) \in (\mu_a \pm T^{-1/4})L^{(1)}$. Further, let $a_*^{(2)} = \arg\min_{a \in \mathcal{A}} \mu_a L^{(1)} - X_a$, then, due to the memorylessness of the Geometric Distribution (see Theorem H.1 for more details), we have that with probability at least $1 - \varepsilon T^{1/4}$

$$\left( \min_{a \in \mathcal{A} \backslash \{a_*^{(2)}\}} \mu_a L^{(1)} - X_a \right) - \left( \mu_{a_*^{(2)}} - X_{a_*^{(2)}} \right) \geq T^{-1/4}.$$

In such a case $a^{(2)} = a_*^{(2)}$. To ensure replicability throughout $t = 1, \ldots, T$, we add appropriate Geometric noise to each expert at the end of each block.

**Worst-Case Regret:** Given the above discussion we know that with high probability the regret of the algorithm never crosses $\approx \log\log(T)\sqrt{T}$ if the cost sequence $S = (c_1, \ldots, c_T)$ is sampled from $\mathcal{D}^{\otimes T}$.

Hence, to achieve a worst-case regret bound, we switch to a usual regret minimizing algorithm such as FTPL (with regret upper bound $\approx \sqrt{T}$), whenever the regret of the algorithm at the current time steps exceeds roughly $\log\log(T)\sqrt{T}$. Now, clearly on any cost sequence $S = (c_1, \ldots, c_T)$ we can never exceed a regret of roughly $\mathcal{O}(\log\log(T)\sqrt{T})$.

Below we formalize the above discussion into Algorithm 9 and Theorem H.1.

---

**Algorithm 9** Follow the Perturbed Leader with varying Noise & Block Lengths

---

**Input:** Cost sequence $S = (c_1, \ldots, c_T)$ arriving one by one over time and replicability parameter $\rho$

**Result:** iid $\rho$-replicable algorithm with worst case regret $K$.

56   $t \leftarrow 0; \alpha \leftarrow \sqrt{\ln(8n \log\log(T)/\rho)}; \gamma \leftarrow \frac{\rho}{8 \log\log(T)}$   // some useful parameters

57   $K \leftarrow 1000 \times \left( \frac{1}{\rho} \times (\log\log(T))^2 \times \log(n \log\log(T)/\rho) \times \sqrt{T} \right)$

58   **for** $i = 1, \ldots,$ **do**

     /* Block $i$ begins                                                                     */

59      $L^{(i)} \leftarrow T^{1-2^{-i}}$   // this is the length of block $i$

60      **if** $t=0$ **then**

61        $a^{(i)} = 1$   // select expert 1 for the first block

62      **else**

63        $\varepsilon^{(i)} \leftarrow \frac{\gamma}{2\alpha t}$   // this is noise parameter used for Block $i$

64

65        $\forall a \in \mathcal{A}$ sample $X_a^{(i)} \sim \mathbf{Geo}(\varepsilon^{(i)})$

66        $a^{(i)} \leftarrow \arg\min_{a \in \mathcal{A}} \left( \sum_{j=1}^{t} c_j(a) \right) - X_a^{(i)}$

67      **for** $u = 1, \ldots, L_i$ **do**

68        $t \leftarrow t + 1$

69        $a_t \leftarrow a^{(i)}$   // select the same expert throughout the block

70        **if** $t = T$ *or* $\mathsf{Reg}_t \geq K - 2\sqrt{T \ln n}$ **then**

71          **go to** Algorithm 9

72   **if** $t < T$ **then**

73      Follow usual FTPL with noise $\mathbf{Geo}(\varepsilon = \sqrt{\ln n/T})$ for the remaining time steps

---

**Theorem H.1** (iid-replicable algorithm for experts). *Algorithm 9 is iid $\rho$-replicable and achieves regret*

$$\mathcal{O}\left( \frac{1}{\rho} \times (\log\log(T))^2 \times \log\left( \frac{n \log\log(T)}{\rho} \right) \times \sqrt{T} \right).$$

*Proof. Regret analysis:* Clearly, the regret of Algorithm 9 can never be more that $K$ because as soon as $\mathsf{Reg}_t \geq K - 2\sqrt{T \ln n}$ we switch to the usual FTPL with geometric noise $\mathbf{Geo}(\varepsilon = \sqrt{\ln n/T})$, which in expectation suffers an additional regret of $2\sqrt{T \ln n}$.

*Replicability analysis:* First, we make a few observations that are helpful to show the replicability property of Algorithm 9. Let $P^{(i)} = \sum_{v=1}^{i} L^{(i)}$, ie, the end of block $i$, and $\mu_a = \mathbb{E}_{c \leftarrow \mathcal{D}}[c(a)]$. We analyze the probability of various events below and the probability should be thought of as over the cost sequence $S \sim \mathcal{D}^{\otimes T}$ and internal randomness $R$.

1. The number of blocks used by Algorithm 9 at Algorithm 9, say $B$, is at most $\log\log(T)+1$. This is because

$$L_{\log\log(T)+1} + L_{\log\log(T)} = T^{1-2^{-\log\log(T)-1}} + T^{1-2^{\log\log(T)}} = \frac{T}{\sqrt{2}} + \frac{T}{2} > T.$$

2. Recall that $\alpha = \sqrt{\ln(8n \log\log(T)/\rho)}$. Let $E_a^{(i)}$ be the event that $\sum_{j=1}^{P^{(i)}} c_j(a) \notin [\mu_a P^{(i)} \pm \alpha\sqrt{P^{(i)}}]$. Hence, if $E_a^{(i)}$ does not take place then the cumulative cost of expert $a$ when block $i$ ends, ie, at time $t = P^{(i)}$, is within $\alpha\sqrt{P^{(i)}}$ of the expected cost of expert $a$. Here, $\sqrt{P_i}$ can be thought of as a proxy for the standard deviation of the cumulative cost up till $P^{(i)}$.

By McDiarmid's inequality (see Theorem B.2), we have $\Pr[E_a^{(i)}] \leq 2\exp(-2\alpha^2)$. Thus,

$$\Pr\left[ E := \bigcup_{\substack{a \in \mathcal{A} \\ i \in [B]}} E_a^{(i)} \right] \leq 2nB \exp(-2\alpha^2) \leq \rho/8.$$

3. Recall that $\varepsilon^{(i)} = \frac{\gamma}{2\alpha\sqrt{P^{(i-1)}}}$ where $\gamma = \frac{\rho}{8\log\log(T)}$. For $i$, let $a_*^{(i)} = \arg\min_{a\in\mathcal{A}}\left(\mu_a P^{(i-1)} - X_a^{(i)}\right)$. Further, let $F^{(i)}$ denote the event that there exists an $a \in \mathcal{A}$ such that

$$\mu_{a_*^{(i)}} P^{(i-1)} - X_{a_*^{(i)}}^{(i)} > \mu_a P^{(i-1)} - X_a^{(i)} - 2\alpha\sqrt{P^{(i-1)}}.$$

Here, we should think of $a_*^{(i)}$ as the expected expert to be selected at the beginning of block $i$ conditioned on the values of $X_a^{(i)}$. When the event $F^{(i)}$ does not happen we have that the expected expert $a_*^{(i)}$ is well separated from other experts such that on typical cost sequences from $\mathcal{D}^{\otimes T}$, ones where $\cup_{a\in\mathcal{A}}E_a^{(i-1)}$ does not occur, it will be the case that $a^{(i)} = a_*^{(i)}$.

We claim that $\Pr[F^{(i)}] \le \varepsilon^{(i)} \times 2\alpha\sqrt{P^{(i-1)}} = \gamma$. To see this, for a fixed action $a \in \mathcal{A}$ and constant $Y \in R$ let $\tilde{F}_{a,Y}$ be the event that $a_*^{(i)} = a$ and

$$\max_{\substack{b\in\mathcal{A}\\b\ne a}}\left(\mu_a P^{(i-1)} - \left(\mu_b P^{(i-1)} - X_b^{(i)}\right)\right) = Y.$$

Upon conditioning, we have

$$\Pr[F^{(i)} \mid \tilde{F}_{a,Y}] = \Pr\left[X_{a_*^{(i)}}^{(i)} < Y + 2\alpha\sqrt{P^{(i-1)}} \mid X_{a_*^{(i)}}^{(i)} > Y\right] \le \varepsilon^{(i)} \times 2\alpha\sqrt{P^{(i-1)}},$$

since $X_{a_*^{(i)}}^{(i)}$ is distributed as $\mathbf{Geo}(\varepsilon^{(i)})$ and using the memorylessness of the Geometric distribution.

Since this is true for all values of $a_*^{(i)}$ and $Y$, we have the above claim. Hence,

$$\Pr\left[F := \bigcup_{i\in[B]} F^{(i)}\right] \le B\gamma < \rho/8.$$

4. Let $\eta = \sqrt{2\ln(16n/\rho)}$ For $a \in \mathcal{A}$, let $G_a$ be the event that there exists a $t \in [T]$ such that $\left|\sum_{j\le t} c_j(a) - \mu_a t\right| > \eta\sqrt{T}$. Hence, when $G_a$ does not occur it means that the cumulative cost of expert $a$ at any time $t$ is within $\eta\sqrt{T}$ of its expected cumulative cost till time $t$. Note that event $E_a^{(i)}$ provides possibly a stronger bound on the cumulative cost of expert $a$ at time $P^{(i)}$, but not at all times. Then, by using Theorem B.1 with parameters $S_t = \sum_{j\le t} c_j(a) - \mu_a t$ and $a = \frac{\eta\sqrt{T}}{2}$, $b = \frac{\eta}{2\sqrt{T}}$, we have $\Pr[G_a] \le 2\exp(-\eta^2/2)$; hence,

$$\Pr\left[G := \bigcup_{a\in\mathcal{A}} G_a\right] \le 2n\exp(-\eta^2/2) < \rho/8.$$

5. Let $\beta = \ln(8n\log\log(T)/\rho)$ and $M^{(i)} = \max\left\{X_a^{(i)} \mid a \in \mathcal{A}\right\}$, and let $H^{(i)}$ be the event that $M^{(i)} > \frac{\beta}{\varepsilon^{(i)}}$. Hence, if $H^{(i)}$ does not occur then the noise added at the beginning of block $i$ to any expert $a \in \mathcal{A}$ is at most a multiplicative factor $\beta$ over the expected value of $1/\varepsilon^{(i)}$.

Then, using the fact that $X_a^{(i)} \sim \mathbf{Geo}(\varepsilon^{(i)})$ and by a union bound over $a \in \mathcal{A}$, we have

$$\Pr\left[H^{(i)}\right] \le n(1 - \varepsilon^{(i)})^{\beta/\varepsilon^{(i)}} \le n\exp(-\beta).$$

Thus,

$$\Pr\left[H := \bigcup_{i\in[B]} H^{(i)}\right] \le nB\exp(-\beta) \le \rho/8.$$

Properties 4 and 5 will be used later to bound the chance of the algorithm calling the usual FTPL (Algorithm 9), and properties 2 and 3 will be used to bound the chance that the same expert is selected across each block in two different runs of the algorithm on iid inputs. In what follows for a RV $Z$ we denote by $Z(S, R)$ the realization of $Z$ on the cost sequence $S \sim \mathcal{D}^{\otimes T}$ and internal randomness $R$: further, for an event $I$ let $I(S, R)$ be 1 if $(S, R) \in I$ and 0 otherwise. To show replicability we need to show that with probability at least $1 - \rho$ (over the internal randomness of the algorithm and over independent cost sequences $S_1, S_2$ from $\mathcal{D}^{\otimes T}$) at all times $t$ we have $a_t(S_1, R) = a_t(S_2, R)$ ($a_t$ is the expert selected by the algorithm at time $t$). For this it is sufficient to show that with probability at least $1 - \rho$: for all $i \in [B]$ we have that $a^{(i)}(S_1, R) = a^{(i)}(S_2, R)$ and Algorithm 9 is never called (i.e., at no time does the regret exceed $K - 2\sqrt{T \ln n}$.)

Now, from the above observations, with probability at least $1 - \rho$ over $R$ and $S_1$ we have that $E(S_1, R) = F(S_1, R) = G(S_1, R) = H(S_1, R) = 0$ and $E(S_2, R) = F(S_2, R) = G(S_2, R) = H(S_2, R) = 0$.

Next, consider any $S_1, S_2, R$ such that $E(S_1, R) = F(S_1, R) = G(S_1, R) = H(S_1, R) = 0$ and $E(S_2, R) = F(S_2, R) = G(S_2, R) = H(S_2, R) = 0$. In such a case we have:

1. $\forall i \in [B] : a^{(i)}(S_1, R) = a^{(i)}(S_2, R)$: This is because $a_*^{(i)}(S_1, R) = a_*^{(i)}(S_1, R)$, since in both cases we are using same randomness $R$. Further, since $E(S_1, R) = F(S_1, R) = 0$ we have that $a^{(i)}(S_1, R) = a_*^{(i)}(S_1, R)$. Similarly, $a^{(i)}(S_2, R) = a_*^{(i)}(S_2, R)$.

2. *At no time does the regret exceed $K - 2\sqrt{T \ln n}$*: Since $G(S_1, R) = H(S_1, R) = 0$, therefore, at any time $t \leq T$ the regret suffered by the algorithm can be analyzed as follows. Let $i + 1$ be the block which contains the time index $t$. Then, for any expert $a \in \mathcal{A}$ at time $t$ the cost incurred on $(S_1, R)$ is:

$$\sum_{j=1}^{t} c_j(a)(S_1, R) \geq \mu_a t - \eta \sqrt{T} \qquad (\because G(S_1, R) = 0)$$

$$= L^{(1)} \mu_a + L^{(2)} \mu_a + \ldots + L^{(i)} \mu_a + (t - P^{(i)}) \mu_a - \eta \sqrt{T}. \qquad (40)$$

The cost incurred by the algorithm till time $t$ is at most:

$$L^{(1)} + L^{(2)} \left( \mu_{a^{(2)}(S_1,R)} + 2\alpha \sqrt{P^{(2)}} \right) + \ldots \quad + L^{(i)} \left( \mu_{a^{(i)}(S_1,R)} + 2\alpha \sqrt{P^{(i)}} \right)$$
$$+ (t - P^{(i)}) \mu_{a^{(i+1)}(S_1,R)} + 2\eta \sqrt{T}. \qquad (41)$$

This is because for a particular block say $v \leq i$ the algorithm selects expert $a^{(v)}(S_1, R)$ and incurs a cost that is close to the expected cost for $a^{(v)}(S_1, R)$ for $L^{(v)}$ time steps: within the $L^{(v)}$ time steps the cost incurred by expert $a^{(v)}(S_1, R)$ can deviate from $\mu_{a^{(v)}(S_1,R)} L^{(v)}$ by at most $\alpha \sqrt{P^{(v-1)}} + \alpha \sqrt{P^{(v)}} \leq 2\alpha \sqrt{P^{(v)}}$ (as $E(S_1, R) = 0$). The additional $2\eta \sqrt{T}$ term accounts for the deviation over the expected cost for the time steps $P^{(i)} + 1, \ldots, t$ (as $G(S_1, R) = 0$).

Now, let us analyze $\mu a^{(v)}(S_1, R) - \mu_a$ for a fixed block $v$. (For ease of presentation in what follows we have removed the explicit reference to $(S_1, R)$ at some places: each RV should be thought of as its realization on $(S_1, R)$.)

Notice that by Algorithm 9 we have:
$$\sum_{j=1}^{P^{(v-1)}} c_j(a^{(v)}) - X_{a^{(v)}}^{(v)} \leq \sum_{j=1}^{P^{(v-1)}} c_j(a) - X_a^{(v)}.$$

Also, as $E(S_1, R) = 0$ we have
$$\mu_{a^{(v)}} P^{(v-1)} - \alpha \sqrt{P^{(v-1)}} \leq \sum_{j=1}^{P^{(v-1)}} c_j(a^{(v)})$$

and

$$\sum_{j=1}^{P^{(v-1)}} c_j(a) \le \mu_a P^{(v-1)} + \alpha \sqrt{P^{(v-1)}},$$

thus, using $X_{a^{(v)}}^{(v)} \le M^{(v)}$, we have

$$\mu_{a^{(v)}} - \mu_a \le \frac{2\alpha\sqrt{P^{(v-1)}} + M^{(v)}}{P^{(v-1)}}.$$

Finally, we can upper bound the regret at time $t$ on $(S_1, R)$ as (Equation (41) - Equation (40)):

$$L_1 + \sum_{v=2}^{B} (\mu_{a^{(v)}} - \mu_a) L^{(v)} + (t - P^{(i)})(\mu_{a^{(i)}} - \mu_a) + 2\alpha B\sqrt{T} + 3\eta\sqrt{T}$$

$$\le L_1 + \sum_{v=2}^{B} L^{(v)} \times \frac{2\alpha\sqrt{P^{(v-1)}} + M^{(v)}}{P^{(v-1)}} + (2\alpha B + 3\eta)\sqrt{T},$$

which, by using $M^{(v)} \le \frac{\beta}{\varepsilon^{(v)}} = \frac{\beta}{\gamma} \times \sqrt{P^{(v-1)}}$ and $P^{(v-1)} \ge L^{(v-1)} = \frac{\left(L^{(v)}\right)^2}{T}$, becomes

$$\le (2\alpha B + 3\eta + 1)\sqrt{T} + (2\alpha + \beta/\gamma)\sum_{v=2}^{B} \sqrt{T}$$

$$\le (2\alpha B + 3\eta + 1)\sqrt{T} + B(2\alpha + \beta/\gamma)\sqrt{T},$$

which upon substituting the values of $\alpha = \sqrt{\ln(8n \log\log(T)/\rho)}$, $\beta = \ln(8n \log\log(T)/\rho)$, $\gamma = \frac{\rho}{8 \log\log(T)}$ and $\eta = \sqrt{2\ln(16n/\rho)}$ becomes,

$$\le 1000 \times \left(\frac{1}{\rho} \times (\log\log(T))^2 \times \log\left(\frac{n \log\log(T)}{\rho}\right) \times \sqrt{T}\right) - 2\sqrt{T \ln n}$$

$$= K - 2\sqrt{T \ln n}.$$

$\square$

# I  Lower Bounds for Replicable Regret Minimization

First we prove a lower bound on the regret of an iid $\rho$-replicable algorithm even with two experts via a reduction from the coin problem of Impagliazzo et al. [2022](Lemma 7.2).

The coin problem is as follows: promised that a $0-1$ coin has bias either $1/2-\tau$ or $1/2+\tau$ for some fixed $\tau > 0$ how many flips are required to identify the coins bias with high probability? Below we state the result from Impagliazzo et al. [2022] regarding the replicability of such an algorithm for the coin problem.

**Lemma I.1** (Impagliazzo et al. [2022](Lemma 7.2): Sample Lower Bound for the coin Problem)**.** *Let $\tau < \frac{1}{4}$ and $\rho < \frac{1}{16}$. Let $\mathcal{B}$ be a $\rho$-replicable algorithm that decides the coin problem with success probability at least $1 - \delta$ for $\delta = \frac{1}{16}$ using $T$ iid samples. Furthermore, assume $\mathcal{B}$ is $\rho$-replicable, even if its samples are drawn from a coin $\mathcal{C}$ with bias in $\left(\frac{1}{2} - \tau, \frac{1}{2} + \tau\right)$. Then $\mathcal{B}$ requires sample complexity $T \in \Omega\left(\frac{1}{\tau^2 \rho^2}\right)$, i.e., $\rho \in \Omega\left(\frac{1}{\tau\sqrt{T}}\right)$.*

Using the above lemma we obtain the following result.

**Theorem I.2** (Lower Bound on the regret of an iid $\rho$-replicable algorithm for two experts)**.** *Suppose there are two experts $\mathcal{A} = \{1, 2\}$, ie, $n = 2$. Let $\rho > 0$ be a parameter. Then, any iid $\rho$-replicable algorithm must suffer a worst-case regret of $K = \min\{\Omega(\sqrt{T}/\rho), T/64\}$ : in fact, there exists a distribution $\mathcal{D}$ over $[0,1]^2$ such that on a cost sequence $S = (c_1, \ldots, c_T)$ sampled from $\mathcal{D}^{\otimes T}$ the expected regret of the algorithm is $K = \min\{\Omega(\sqrt{T}/\rho), T/64\}$.*

**Remark.** *Using the usual regret lower bound for $n$ experts we can conclude a lower bound of $\Omega(\sqrt{T}/\rho + \sqrt{T \log(n)})$ provided this is lesser than $T/64$.*

*Proof.* Let ALG be an iid $\rho$-replicable algorithm with worst-case regret $K$. Further, let the expert selected by ALG at time $t$ be $a_t$. We reduce an instance of the coin problem from above with parameters $\rho$, $\tau = \frac{K}{\delta T}$, and $\delta = 1/16$ to the expert setting as follows. For a sequence of coin flips $\hat{c}_1, \hat{c}_2, \ldots, \hat{c}_T$, where each $\hat{c}_j \in \{0, 1\}$, let $c_1, \ldots, c_T$ be the cost sequence such that $c_j(1) = \hat{c}_j$ and $c_j(2) = 1 - \hat{c}_j$.

Now, we feed ALG the cost sequence $c_1, \ldots, c_T$ corresponding to iid coin flips $c_1, c_2, \ldots, c_T$ where the bias of the coin is between $1/2 + \tau$ and $1/2 - \tau$. For such inputs we know that ALG is $\rho$-replicable. Further, when the coin's bias is either $1/2 + \tau$ or $1/2 - \tau$, $a = \text{MAJ}\{a_t \mid t \in [T]\}$ is a good indicator of the coin's bias: we answer that the coin has bias $1/2 + \tau$ if $a = 2$ and $1/2 - \tau$ if $a = 1$. Also, we will be correct with probability at least $1 - \delta$. To see this, suppose that the coin has bias $1/2 + \tau$ and notice that the expected cost of ALG is (the expectations below are over the internal randomness of the algorithm and the coin flips leading to the sequence $c_1, \ldots, c_T$):

$$\mathbb{E}[\text{Cost(ALG)}] = \mathbb{E}[\sum_{j=1}^{T} \mathbb{I}[a_t = 1]c_t(1) + \mathbb{I}[a_t = 2]c_t(2)]$$

$$= \sum_{j=1}^{T} \mathbb{E}[\mathbb{I}[a_t = 1]c_t(1) + \mathbb{I}[a_t = 2]c_t(2)] \qquad \text{(Linearity of Expectation)}$$

$$= \sum_{j=1}^{T} \mathbb{E}[\mathbb{I}[a_t = 1](1/2 + \tau) + \mathbb{I}[a_t = 2](1/2 - \tau)] \qquad (\because a_t \text{ is independent of } c_t)$$

$$= T/2 - \tau \, \mathbb{E}\left[\left(\sum_{j=1}^{T} \mathbb{I}[a_t = 2] - \mathbb{I}[a_t = 1]\right)\right].$$

Using the fact that the regret is at most $K$ and that $\mathbb{E}[\sum_{j=1}^{T} c_j(2)] = T/2 - \tau T$ (where the expectation is over the coin flips) we get:

$$T/2 - \tau \, \mathbb{E}\left[\left(\sum_{j=1}^{T} \mathbb{I}[a_t = 2] - \mathbb{I}[a_t = 1]\right)\right] \leq T/2 - \tau T + K$$

$$\implies \mathbb{E}\left[\left(\sum_{j=1}^{T} \mathbb{I}[a_t = 2] - \mathbb{I}[a_t = 1]\right)\right] \geq T - K/\tau = (1 - \delta)T \quad \text{(using } \tau = \frac{K}{\delta T})$$

$$\implies \Pr[\text{MAJ}\{a_t \mid t \in [T]\} = 2] \times T \geq (1 - \delta)T$$

$$\implies \Pr[\text{MAJ}\{a_t \mid t \in [T]\} = 2] > 1 - \delta.$$

Hence, we can use Lemma I.1 to conclude that $\rho > \Omega(\frac{1}{\tau\sqrt{T}})$ provided $\tau < 1/4$. Using $\tau = \frac{K}{\delta T}$ and $\delta = 1/16$ the constraint on $\tau$ translates to $K < T/64$. This gives the theorem.

$\square$

Now, we use the above lemma to obtain the lower bound on regret in the case of $n \geq 2$ experts when we have an adversarially $\rho$-replicable algorithm.

**Theorem I.3** (Lower Bound on the regret of an adversarial $\rho$-replicable algorithm for $n$ experts)**.** *Suppose there are $n$ experts $\mathcal{A} = \{1, 2, \ldots, n\}$. Let $\rho > 0$ be a parameter. Then, any adversarially $\rho$-replicable algorithm must suffer a worst-case regret of $\min\{\Omega(\sqrt{T \log(n)}/\rho), T/64\}$.*

*Proof.* Let the set of experts be $\mathcal{A} = \{1, 2, \ldots, n\}$. It is sufficient to prove the above theorem when $n = 2^m$ for $m \in \mathbb{N}$. We will proceed by induction to prove that the worst-case regret for any iid $\rho$-replicable algorithm, say ALG, is at least $\min\{U \times \frac{\sqrt{Tm}}{\rho}, T/64\}$ where $U > 0$ is a universal constant.

For $m = 1$ this follows from Theorem I.2. For $m > 1$ we proceed as follows. WLOG we can assume $\rho$ is such that $U \times \frac{\sqrt{Tm}}{\rho} \leq T/64$ or else we can always increase the value of $\rho$ until the inequality is satisfied. Let $L = T/m$ and consider a cost sequence $\hat{c}_1, \ldots, \hat{c}_L$ for two experts $\{1, 2\}$, ie, each $\hat{c}_j \in [0, 1]^2$. Let $c_1, \ldots, c_L$ be the cost sequence for $n$ experts such that $c_j(a) = \hat{c}_j(1)$ for $a \in [n/2]$ and $c_j(a) = \hat{c}_j(2)$ for $a > n/2$.

By using Theorem I.2 we know that there exists a sequence of $\hat{c}_1, \ldots, \hat{c}_L$ such that the regret of ALG on $c_1, \ldots, c_L$ is at least $\min\{U \times \sqrt{L}/\rho, L/64\} = U \times \sqrt{L}/\rho$. Further, either all experts $1, 2, \ldots, n/2$ or all experts $n/2 + 1, \ldots, n$ are the min-cost experts for $c_1, \ldots, c_L$. WLOG let us assume that it is the former case.

Now, for a cost sequence $\hat{c}_{L+1}, \ldots, \hat{c}_T$ for the experts $1, 2, \ldots, n/2$, ie, each $\hat{c}_j \in [0, 1]^{n/2}$, let $c_{L+1}, \ldots, c_T$ be the cost sequence such that $c_j(a) = \hat{c}_j(a)$ if $a \in [n/2]$ and $c_j(a) = 1$ if $a > n/2$. Notice that even conditioned on receiving $c_1, \ldots, c_L$, ALG remain adversarially $\rho$-replicable. By induction we can find a cost sequence $\hat{c}_{L+1}, \ldots, \hat{c}_T$ for the $n/2$ experts $1, 2, \ldots, n/2$ such the regret of ALG (conditioned on $c_1, \ldots, c_L$) on $c_{L+1}, \ldots, c_T$ is at least $\min\{U \times \frac{\sqrt{(T-L)(m-1)}}{\rho}, \frac{T-L}{64}\} = U \times \frac{\sqrt{(T-L)(m-1)}}{\rho}$. Hence, the overall regret of ALG is

$$U \times \sqrt{L}/\rho + U \times \frac{\sqrt{(T-L)(m-1)}}{\rho} = U \times \sqrt{Tm}/\rho.$$

This can be seen by substituting $L = T/m$.

$\square$

