# OpenReview forum: "Replicable Online Learning"
_NeurIPS.cc/2025/Conference — NeurIPS 2025 poster_

### Official Review · Reviewer_GsvK · 2025-06-15

**Clarity:** 2
**Significance:** 3
**Originality:** 3
**Rating:** 4
**Confidence:** 3

**Summary:**

This paper studies algorithmic replicability in the online setting. In this setting, an oblivious adversary picks a sequence of distributions over loss vectors before the game begins. The goal is to design low-regret online algorithms that, with high probability, output the exact same sequence of actions on two independently sampled sequences of loss vectors. The authors consider two online learning settings —online linear optimization and learning with expert advice —and design replicable online algorithms that achieve sublinear regret. In the special case where the adversary chooses the same loss distributions for all rounds, the authors design a replicable online learning algorithm with near-optimal regret guarantees. More generally, the authors present a generic conversion of a non-replicable online algorithm into a replicable one. Finally, the authors complement their upper bounds with lower bounds on regret that any replicable algorithm needs to suffer.

**Questions:**

It would be great if the authors could address all my concerns in Weaknesses 1-4.

Additional questions:
1. The technique of batching (and more generally, lazy algorithms) has been studied extensively in online learning (see [1]). What implications do these results have on replicable online learning, if any?

2. Lazy online algorithms have been especially useful for designing differentially private online learning algorithms (see [2]). How does your notion of replicable online learning relate to differentially private online learning? Does one imply the other?

3. Learning with expert advice is a special case of online learning optimization when the decision space is taken to be the n-dimensional simplex. Can your regret bound for replicable online linear optimization be converted into a regret bound for learning with expert advice? That is, do you really need to have a separate algorithm for the problem of replicable learning with expert advice?

[1] Arora, Raman, Ofer Dekel, and Ambuj Tewari. "Online bandit learning against an adaptive adversary: from regret to policy regret." arXiv preprint arXiv:1206.6400 (2012).

[2] Asi, Hilal, et al. "Private online learning via lazy algorithms." Advances in Neural Information Processing Systems 37 (2024): 112158-112183.

**Ethical Concerns:**

["NO or VERY MINOR ethics concerns only"]

**Final Justification:**

The authors addressed my main concerns and provided a concrete plan for improving the readability and exposition in the camera-ready version. As such, I have increased my score from a 3 to a 4.

**Limitations:**

There is a significant gap between the regret upper and lower bounds. It would be great if the authors could provide some intuition on how this gap can be reduced.

**Paper Formatting Concerns:**

No formatting concerns

**Quality:**

2

**Strengths And Weaknesses:**

Strengths:
- The paper studies an important problem in machine learning and presents, to the best of my knowledge, the first formal framework for studying replicability for online learning with full-information feedback.
- The authors provide several interesting results across various online learning settings, including both lower and upper bounds

Weaknesses: My main concern with this paper is its presentation, which is the main reason behind my rating. Below, I list my concerns.
1. There are several places where notation is not well-defined and hinders readability:
  - In many places (Line 248, 274, ...), vectors are being added to spaces. This operation should be defined.
  - In Line 272, $Z^n$ is not defined.
  - In Line 274, $c_{1:t-1}$ is not defined.
  - In Theorem 3.1, $D$ is not defined.
  - Figure 1 is a bit unclear, it would be good if the text c_{1:t-1} was colored in red, and c'_{1:t-1} is colored blue
  - In Algorithm 2 and Line 287, "Geo" and "geometric noise" are not defined. What does geometric noise mean? What is the distribution of this noise? This should be stated precisely in the main text.
  - In Line 316, I think $\Omega$ should be $O$
  - In Line 312, $D$ is not defined.
  - In Line 326, $K$ is not defined. Also, what am I supposed to take away from this sentence? There is always an upper bound on the regret...

2. The paper currently reads like a list of results without providing much intuition. No intuition/sketches behind the proofs is provided for any of the main Theorems, as the authors repeatedly "defer the proof to the Appendix." The most egregious offense is Section 4, which contains just the algorithm and the theorem statement. The authors should expand on this section and provide insight on why geometric noise is needed and why the results for online learning optimization don't just apply to the expert setting (see Question 3 in "Additional Questions"). Section 5 is also not illustrative as Algorithm 5 is not in the main text. The authors should move Algorithm 5 into the main text and provide a proof sketch. Overall, more exposition is needed to ensure that the main text does not just read like a list of results, and hence, I believe that this work requires a rewrite.

3. A minor concern is the significant gap between the lower and upper bounds on the regret for replicable online linear optimization. It would be good if the authors could comment on whether a tighter analysis of the current approach can lead to a better upper bound than the one in Theorem 3.1, and it not, what the barrier is.

4. Another minor concern is regarding novelty. It seems like the techniques used by this paper are not new: rounding has been used by existing work on replicability, and "blocking" (more generally, laziness) has also been studied in online learning literature. It would be great if the authors could comment on what makes this work not just a combination of these two techniques.

---

> ### Author Rebuttal · Authors · 2025-07-30
>
> We thank the reviewer for their time and feedbacks, here is our response:
>
> weakness1:
>
> 1.1 - Sure, we will clarify this in the text. It means the cube starting from the point $\sum_{i=1}^{t-1} c_i$.
>
> 1.2 - We will clarify this. Here, we mean a grid with side $1/\epsilon$ and a starting point of $p$.
>
> 1.3 - We will clarify this. This means $\sum_{i=1}^{t-1} c_i$.
>
> 1.4 - D is the diameter of the action set (pointed in Thm D.3), we will add this to the model.
>
> 1.6 - We will add this to the model section. The geometric distribution with parameter $\epsilon$ is the discrete probability distribution that describes when the first success in an infinite sequence of independent and identically distributed Bernoulli trials occurs. Its probability mass function depends on epsilon and is defined as:
> $P(X=k)=(1−\epsilon)^{k−1}\epsilon$
>
> 1.7 - You are correct. We will fix that.
>
>
> 1.9 - Apologies for the confusion: what we mean is the following. The goal is to design an iid $\rho$-replicable algorithm with worst-case regret as low as possible. Recall the following
> from Section 2: (1)We say an algorithm has Worst-case regret $K$ if  for any cost sequence $S = (c_1, . . . , c_T )$, the expected regret satisfies $\E_{R\sim \cal{R}}[Reg(S,R)]$ (2)We say an algorithm has replicability $\rho$ if for iid cost sequence.
>
>
> weakness 2:
> Due to the space issues, our remedy for this was to have a longer introduction with an extensive “our contributions” section to explain the intuition behind our results.
>
> weakness 3:
>  Regarding the upper bound, our current algorithm would not achieve a better bound. The reason is that, there are two variables in our algorithms, one the block size B, and the other one, the noise level $\epsilon$. In order to bound the probability of non-replicability to at most $\rho$, we need to set these two parameters carefully (equation below the line 853). Furthermore, in order to minimize regret, we need to set $\epsilon = $\frac{1}{\sqrt{BT}}$, line 856. Putting these not inequalities together, gives us the optimum values of noise level epsilon and block size B. However, there might be a different algorithm that achieves a better regret bound, and we leave it as an open question.
>
> weakness 4:
> Our main contribution is a principled framework to study replicability with low regret in settings combining stochastic and adversarial elements, whereas previous works focused exclusively on the stochastic case. By considering an adversarial (oblivious) \emph{sequence of distributions}, our model generalizes the standard oblivious adversary (point-mass distributions) and introduces a natural approach to replicability in adversarial environments.
> In hindsight, although the employed techniques are standard, it was not a-priori clear that our definitions would yield sublinear regret. Furthermore, the analysis presents technical challenges: certain scenarios do not admit the ideal $\sqrt{T}$ regret bound (even when $\rho$, the replicability parameter, is constant), whereas others (e.g., i.i.d. experts) achieve favorable bounds.
>
> Question 1:
> We are borrowing from previous ideas but making them work for replicability requires delicate balance and some cleverness to circumvent certain bottlenecks. We study replicability with low regret in settings combining stochastic and adversarial elements, whereas previous works focused exclusively on the stochastic case. By considering an adversarial (oblivious) \emph{sequence of distributions}, our model generalizes the standard oblivious adversary (point-mass distributions) and introduces a natural approach to replicability in adversarial environments.
> In hindsight, although the employed techniques are standard, it was not a-priori clear that our definitions would yield sublinear regret. Furthermore, the analysis presents technical challenges: certain scenarios do not admit the ideal $\sqrt{T}$ regret bound (even when $\rho$, the replicability parameter, is constant), whereas others (e.g., i.i.d. experts) achieve favorable bounds.
>
> Question 2:
> connections between online replicability and DP online learning:
>
> We don't see any direct reduction. For example, we require the entire sequence to be replicated whp, not just any given entry, so it's not at all clear if differential privacy with meaningful parameters would be useful for replicability.  In the reverse direction, we should be able to get privacy for a single entry changing by a bounded amount, but full replicability might be an overkill for that.
>
> Question 3:
> Our results for learning from experts is in Section 4 and Theorem 4.1. That being said, by using geometric noise, we end up getting a better dependency on n(number of experts) that is logarithmic instead of polynomial dependency(n^{1/6}) on n for online linear optimization.

---

> ### Comment · Reviewer_GsvK · 2025-08-01
>
> I thank the authors for their response and clarifying the polynomial dependence on the number of experts for OLO . However, I am still concerned about the overall presentation of the paper. I will increase my score from 3 to 4 contingent on the authors  explaining how they plan to address the presentation issues highlighted in Weakness 2 (i.e improving the exposition, intuition, and proof sketches around their main results)

---

> ### Author Response · Authors · 2025-08-01
>
> We thank the reviewer for their response. We understand your point and here is how we plan to improve the presentation of the paper:
>
> Since we have a longer "our contributions" section that goes through the intuition of all the results, we can just keep the results corresponding to Section 6 and 7 in the Apendix. This would be ok since Sections 1.1.4 and 1.1.5 are already giving the intuition behind these results in the main body.
>
> Then we would use this freed-up space to explain our results in Section 3 and 4 (adversarially $\rho$-replicable online linear optimization and experts) more thoroughly. That being said, after Thm 3.1. we would add the following paragraph (similar to lines 757-766 in the current version):
>
> "First, we analyze the regret of Algorithm 1.
> However, we need to be careful when picking the values of $\epsilon$ and $B$ so that the regret does not blow up. If $\epsilon$ is too large then in Algorithm 1, a lot of different accumulated cost vectors $c_{1:t-1}$ get mapped to the same grid point $g_{t-1}$ which would cause a lot of regret. Similarly, when $B$ is too large, the algorithm takes the same action for a large block of time and does not update its decision which would cause the regret to suffer. In order to pick optimal values for $\epsilon$ and $B$, we analyze the regret of Algorithm 1 by appealing to a different algorithm ''Follow the Perturbed Leader with Block Updates'' ($FTPLB(\epsilon,B)$) that in expectation behaves identically to $FLLB(\epsilon,B)$ on any single period and incurs the same expected cost. This algorithm is a modified version of the ``Follow the Perturbed Leader'' ($FTPL(\epsilon)$) algorithm by Hannan-Kalai-Vempala. "
>
> Then we would add a summary of FTPLB algorithms that would be similar to lines 766-775. Also, add lines 776-781 that explain why FTPLB and FLLB have the same expected cost. After that, we present the regret bounds for FTPLB algorithm that would be moving Thm D.3 to the main body (keep the proof in the appendix). After that, we are ready to present a proofsketch for Thm 3.1. This would explain the replicability part and how to put the bounds for replicability and regret together to derive optimal values for $\epsilon$ and $B$ and the final regret bound.
>
> For Section 4, we will explain how to prove Thm 4.1. That is, first we analyze its regret, which would be moving Thm E.1. lines 862-863 to the main body. Adding a proof-sketch for it would be easy, we just need to state that proof uses the following points: 1-the difference between the cost of FTPLB* and BTPL is bounded. 2- the difference between the cost of BTPL and the best expert in hindsight is bounded. Then we add a proof-skecth for Thm 4.1. The prooksketch would be to first explain the replicability bound (similar to lines 944-947) and then explaining how we put the bounds for regret and replicability together to find optimal values for $\epsilon$ and $B$. Furthermore, we will also add a remark to highlight why adding geometric noise derives better regret bounds in the experts setting compared to online linear optimization.
>
> This re-write would give a better presentation of Section 3 and 4., it would also perhaps allow us to move Alg 5 to the main body. As we said earlier, moving Sections 6 and 7 would be alright since the main intuition behind these results are already explained in the our contributions section.
>
> Happy to incorporate any further suggestions, and thanks for your feedback.

---

> > ### Comment · Reviewer_GsvK · 2025-08-03
> >
> > I thank the authors for the detailed plan. As promised, I have increased my score from 3 to 4.

---

> > > ### Author Response · Authors · 2025-08-04
> > >
> > > Thank you for the insightful questions and feedback, we will update the manuscript to reflect all these points.

---

### Official Review · Reviewer_JwSX · 2025-06-26

**Clarity:** 1
**Significance:** 2
**Originality:** 3
**Rating:** 3
**Confidence:** 4

**Summary:**

This paper studies the recently-introduced notion of replicability in online learning setups. An adversary decides on a sequence of $T$ (possibly different) distributions, from which the instances are drawn independently. The online learning algorithm is called *replicable* if, fixing the randomness in the algorithm, the outputs match with high probability on two sequences of instances sampled independently. The goal is to design low-regret and replicable algorithms for various online learning settings.

In comparison, prior work has studied replicable online learning in more restricted settings and/or with weaker guarantees:
- Some focused on the special case that all $T$ distributions are identical (termed iid-replicability in this work).
- Some only requires the algorithm to be low-regret on observations drawn from the distributions (rather than on all sequences).

The main results of this work are summarized below, where polynomial dependence on other parameters (including the replicability parameter) are suppressed:
- Theorem 3.1: A $T^{5/6}n^{1/6}$ regret upper bound for online linear optimization in $n$ dimensions.
- Theorem 4.1: A $T^{5/6}\log^{5/6} n$ regret upper bound for the experts problem with $n$ experts.
- Theorem 5.1: A general transformation that translates a low-regret (but not necessarily repliable) algorithm into a replicable one.
- Theorem 6.1: A $\sqrt{T}\log(n) / \rho$ regret upper bound (modulo a $\mathrm{polylog}(\log T, 1 / \rho)$ factor) for the experts problem in the iid replicable setting.
- Theorems 7.1 and 7.2: Regret lower bounds against replicable online learners in the experts problem.

**Questions:**

Repeating the "Weaknesses" section above:
- In the online setting, why is it reasonable to assume that one can "repeat the history" and observe a sequence of identically distributed data points?
- For the proof of Theorem 5.1, could you clarify whether the proof relies on some unstated assumptions on the loss, as well as why the step from (14) to (15) is valid?

**Ethical Concerns:**

["NO or VERY MINOR ethics concerns only"]

**Final Justification:**

During the discussion, the authors acknowledged that one of the main results (Theorem 5.1) holds for a more restricted family of problems---namely, those with a linear loss function $f$---than what is stated in the original submission, and that its proof needs to be slightly revised for it to be rigorous.

Although the proposed revision makes sense to me at a high level, unfortunately, its validity and rigor could not be fully evaluated (without reading a fully-revised version of the paper). As pointed out in the other reviews, the technical writing is hard to follow at times due to multiple inaccuracies and undefined notations. Therefore, while this paper studies an interesting problem and makes solid progress, I was afraid that it would be risky to accept this manuscript as is, and thus would still lean towards rejection.

**Limitations:**

Yes

**Quality:**

2

**Strengths And Weaknesses:**

Strengths:
- The formulation of replicable online learning is natural and clean from a theoretical perspective. It is stronger (and arguably more natural) than those in prior work. Yet, a sublinear regret bound can still be guaranteed for several fundamental online learning settings.
- The writing is generally clear and easy to follow. Especially, the authors did a good job in comparing their model to prior ones and highlighting the differences.

Weaknesses:
- I was a bit confused about the statement of Theorem 1.3 (and Theorem 5.1): What families of online learning problems does this result apply to? As stated, there isn't any restriction other than that each instance $c_t$ is a $n$-dimensional vector.
- However, as I followed the proof of Theorem 5.1 in Appendix F, it appeared to me that more assumptions are needed. For example, in the step from Equation (13) to (14), isn't it assumed that the loss function $\mathsf{Cost}(\cdot, \cdot)$ is linear in the second argument?
- I didn't follow the step from (14) to (15) either: even assuming the aforementioned linearity, $g_{uB}$ and $g_{(u-1)B}$ are $\sum_{t=1}^{uB}c_t$ and $\sum_{t=1}^{(u-1)B}c_t$ rounded to two different grids---why is the difference between them equal to $c_{(u-1)B:uB}$?
- Except for the iid-replicable experts setting, the upper bounds do not come with lower bounds with matching dependences on $T$.
- From a more practical perspective, I didn't find that the notion of replicability makes as much sense in the online setting as in the batch/offline/supervised setting. In supervised learning, it makes sense to collect fresh data and replicate the learning. In the online setting, why is it reasonable to assume that one can "repeat the history" and observe a sequence of identically distributed data points?

Other comments / writing suggestions:
- Line 63: There is an extra space before the period.
- Line 81: The term "adversarially $\rho$-replicable" was a bit misleading to me: compared to being "iid $\rho$-replicable", the only change seems to be that the observations can be differently distributed across timesteps. In other words, they are still independently sampled from distributions, and I didn't quite see what is "adversarial" in this formulation.
- Appendix F, Line 27 of Algorithm 5: Should the condition be "$t - 1$ is a multiple of $B$ and $t > 1$"? This is because we shouldn't call the internal algorithm at time $t = 1$.

Overall, while I liked the formulation of the problem and the suite of results that the authors obtained, I have strong concerns over the validity (or at least the scope of applicability) of one of the main results. Therefore, I don't think this submission should be accepted without adequately addressing this point.

---

> ### Author Rebuttal · Authors · 2025-07-30
>
> We thank the reviewer for the feedback and reviewing efforts, here is our response:
>
> Weakness 1:
>
> Yes, thank you for pointing this out: we forgot to mention this condition in the theorem statement. We will include the condition in the theorem statement. The reason is that we need the loss $f (a, c)$ to scale linearly with the `1-norm with c and that if $c'$ is a
> random variable with $E[c'] = c$ then $E[f (a, c')] < f (a, c)$ (concavity). But we want to highlight that even with
> the constraint of linearity of the loss function we are able to capture many settings such as online linear optimization, experts and bandit problems.
>
> weakness 2,3:
> We need to make a slight change to the following in Algorithm 5: $g_t \gets G \cap \\{c_{1:t-1} + (-\epsilon/2,\epsilon/2)^n\\}$ so that over the randomness of the initial random offset of the grid G we have $E_[g_t] = c_{1:t-1}$.
> Similarly, for G'. We will make this change: it is a minor one and not conceptual.
>
>
> In light of the previous point, consider the following: in Equation (13), let us bring the $E_{R_{ext}}$ inside. Then we get an expression like:
> $E_{R_{int}} [ E_{R_{ext}} [Cost(a_u, g_{uB} - g'_{(u-1)B})] ]$
>
> which by our assumption we can say is at most:
> $E_{R_{int}} [ Cost(a_u, E_{R_{ext}}[g_{uB} - g'_{(u-1)B} \mid a_u]) ].$
>
> And now, using the fact that $a_u$ is **independent** of both $g'_{(u-1)B}$ and
>
> $g_{uB}$, we have:
>
> $E_{R_{ext}}[g_{uB} - g'_{(u-1)B} \mid a_u]$
>
> $= E_{R_{ext}}[g_{uB} - g'_{(u-1)B}]$
>
> $= c_{uB} - c_{(u-1)B}$
>
> $= c_{(u-1)B:uB}$
>
> This is the case since at a transition point $t$ (that is when $t = (u-1)B + 1$ and we are permitted to change our action), $a_t$ depends on $g_{t-1}$ and $g'_{(t-1-B)}$,
>
> implying that $a_u$ depends on $g_{(u-1)B}$ and $g'_{(u-2)B}$
>
>  (and not $g'_{(u-1)B}$
>
> and $g_{uB}$)). Please see line 33 in Alg 5
>
> In fact, this independence from $g'_{(u-1)B}$ was the main reason for using *two different grids*. Also, due to the rounding procedure we have that
>
> $E[g_{uB}] = c_{uB}$
>
> similarly for g'.
>
>
> weakness 5:
> Here is an example in the online setting: Suppose that we have an algorithm for deciding which routes to use to drive to work each day.  We run it for T days, jotting down features of that day.  For instance, Day 1 was Monday at 8am and sunny.  Day 2 was Tuesday at 10am and rainy, etc.  If you imagine that traffic at any given time is a probabilistic function of the features, then what our guarantees say is that if you run it again on a different sequence of T days then (a) we'll have sublinear regret no matter what, and (b) if this new sequence has the same features as the first sequence, then with high probability your entire sequence of actions will be the same.
>
> other comments,
>
> line 81: In the adversarial case, the distributions across different timesteps are picked by an adversary. That’s why we call it adversarial $\rho$-replicable case.
>
> Appendix F, Sure, we will edit the text, thanks.

---

> > ### Comment · Reviewer_JwSX · 2025-08-01
> >
> > Thank you for the responses! The suggested fix (regarding Weaknesses 2 and 3) makes sense to me. I have a few follow-up questions on your response to Weakness 1:
> >
> > - I didn't exactly follow the claim that it suffices to assume that $f(a, c)$ scales linearly in the $1$-norm of $c$ and that $f$ is concave in $c$. How is the first property formally defined, and where is it used in your proof?
> >
> > - What is an example of $f$ that satisfies the two properties but is non-linear?
> >
> > - For the concavity part, I followed that it helps upper bounding the total cost of the algorithm (i.e., going from (13) to (15)). What I didn't follow was, when lower bounding the cost of the optimal action later in the proof (i.e., going from (19) to (20)), don't we need the loss to be **convex** to move the expectation into the loss? If so, is linearity necessary after all?
> >
> > - Regarding your statement that linear losses already capture settings like OLO, the experts problem, and bandits: Is it true that Theorem 5.1, when combined with the regret bound of the experts problem, also gives a slower $T^{7/8}$ rate (as in Corollary 5.2) compared to the $T^{5/6}$ rate obtained by the specialized algorithm in 1.2?
> >
> > - Furthermore, I didn't quite follow how Theorem 5.1 could apply to the bandits problem, given that the formulation in the paper assumes full feedback rather than bandit feedback.

---

> > > ### Author Response · Authors · 2025-08-04
> > >
> > > We thank the reviewer for their detailed reading and insightful questions. We address each point below.
> > >
> > > ### Points 1, 2, and 3: Clarification on the Linearity Assumption
> > >
> > > We apologize for the lack of clarity in our previous explanation. To clarify, we will explicitly add the assumption that the function $f(a,c)$ is **linear** in the cost vector $c$. Formally, this means $f(a, c_1 + c_2) = f(a, c_1) + f(a, c_2)$ for any cost vectors $c_1$ and $c_2$.
> > >
> > > What we wanted to say was that the derivation from (13) to (15) only requires concavity of $f$ (specifically, $E[f(a,c')] \leq f(a,c)$ where $E[c'|a] = c$). However, the stronger assumption of linearity is necessary for other parts of our proof, such as the normalization of cost vectors supplied to the internal algorithm. It turns out that concavity would also suffice for the step from (19) to (20) since the inequalities point in the desired direction, the necessity of linearity elsewhere in the analysis makes this point moot.
> > >
> > > ### Point 4: Regret Bound for General Algorithms
> > >
> > > Yes, this is correct. We discuss this point in **Appendix G (Corollary G.3)**. The resulting regret bound is worse because our framework is designed to accommodate **any** internal regret-minimizing algorithm (e.g., Multiplicative Weights), not just those based on a Follow the Perturbed Leader (FTPL) template. Our general algorithms (Algorithms 5 and 6) are applicable in this broader, more general context, which comes at the cost of a looser bound.
> > >
> > > ### Point 5: Application to the Multi-Armed Bandit Problem
> > >
> > > We will add a discussion of the bandit problem to the final version of the paper. Our approach uses a standard reduction from the bandit problem to the experts setting (e.g., Chapter 4 of Algorithmic Game Theory, Nisan et al., eds.), which is effective when the number of arms $N$ is much smaller than the number of time steps $T$.
> > >
> > > This method yields a regret bound that is sub-linear in $T$ but suboptimal compared to classical MAB algorithms. The reduction proceeds as follows:
> > > 1.  The $T$ time steps are grouped into $T/B$ buckets, each of size $B$.
> > > 2.  Within each bucket, we pull each of the $N$ arms at $N$ randomly sampled time steps, one for each arm.
> > > 3.  The observed cost for each arm serves as an estimate of its average cost for that bucket. This process generates an $N$-dimensional cost vector for each bucket, effectively converting the problem into a full-information setting over $T/B$ steps.
> > >
> > > This standard reduction leads to a sub-linear regret of $O(T^{2/3})$, ignoring dependencies on $N$.
> > >
> > > To make this reduction replicable, we apply the grid-snapping technique from **Theorem G.2**. Instead of feeding the sampled cost vector to the internal algorithm, we supply the nearest grid point. This modification maintains a sub-linear regret bound, though with a slightly worse dependence on $T$ than in the reduction.

---

> > > > ### Comment · Reviewer_JwSX · 2025-08-04
> > > >
> > > > Thank you for your detailed reply! They make sense to me and have addressed all my concerns. I will update my evaluation accordingly, and would encourage the authors to revise the manuscript based on the discussion.

---

> > > > > ### Author Response · Authors · 2025-08-04
> > > > >
> > > > > Thank you for the insightful questions and feedback. We will update the manuscript to reflect all these points.

---

### Official Review · Reviewer_Pyha · 2025-06-28

**Clarity:** 4
**Significance:** 4
**Originality:** 4
**Rating:** 5
**Confidence:** 4

**Summary:**

\paragraph{Summary} The paper studies online learning problems, like learning with experts and online linear regression under the requirement of replicability. Loosely speaking, a randomized  algorithm on input sampled from some distribution, is replicable if for two independent set of samples, it is very likely over the internal randomness of the algorithm (probability $\geq \rho$ for some replicability parameter $\rho$) that the algorithm returns the same output on the two sets of samples. The notion was introduced motivated by the replicability crisis in science, but the property is interesting as a stability-notion in its own right, and has connections to concepts like differential privacy and generalization.

The main results of the paper are replicable algorithms for online learning with experts and online linear optimization (plus a more general framework for online learning problems) with sublinear regret. The input in each round $i$ is assumed to be sampled from some distribution $\mathcal{D}_i$ chosen by an adversary in an oblivious fashion which significantly generalizes a framework by Esfandiari et al. [2022] where the input is assumed to be iid. The paper also proves almost matching lower bounds for the iid setting through a reduction from replicable coin testing.

The main algorithmic idea is to partition the sequence of updates into blocks (for the  iid setting the blocks have increasing lengths) where the algorithm within each block plays the same action. Long blocks of course incurs worse regret bounds but they are good for replicability where we only have to worry about the ends of blocks. In order to enforce replicability by the end of the block, their algorithm rounds the regularized cost vector to a random grid point and uses the minimum of that grid points as the action for the next block. In order to not make the grid too big, this requires some concentration bounds for the total cost incurred during a block. The final algorithm chooses the block size appropriately to ensure replicability and low regret.

**Questions:**

l74: It would be nice with a discussion of whether obliviousness is necessary. I had a vague intuition that since the algorithms are replicable, an adaptive adversary wouldn't be much harder to handle since over the two independent runs, replicability ensures that the outputs are likely the same and then it seems like the adversary adaptively chosing the distribution shouldn't help much. But it's very possible I'm missing something.

l117-118: I'm confused. Doesn't your algorithms have no switching within the blocks?

l209-214: Don't you need some assumption on the norm of the cost vectors? Is it related to the $D$ on l279 (which I think hasn't been defined).

Example 3.1. I had a hard time following the writing in this example and I don't know the precise meaning of "gives a bonus" or "pretends that $a_2$ has better performance initially". The final sentence could also be phrased stronger I think.

l784: In the def of $a_t$ there is a missing $+$.

**Ethical Concerns:**

["NO or VERY MINOR ethics concerns only"]

**Final Justification:**

Please see my discussion with the authors

**Limitations:**

Yes

**Quality:**

4

**Strengths And Weaknesses:**

\textbf{Strengths}:
(1) I found the problem well-motivated and their model very interesting and natural opening up for several new directions for future work.
(2) The technique of partitioning into blocks and using random grid shifting after the blocks to enforce both replicability and low regret is very nice.
(3) It is great that the paper provides a near matching lower bound in the iid setting
(4) The paper is well-written, and I had little problems following the proofs. I read the first 9 pages and the proof of theorem 3.1 in the appendix, but stopped after that.
(5) The paper states many interesting problems for future work.

\textbf{Weaknesses} (1) The paper might benefit by a discussion of how the techniques differ from past work, e.g., Esfandiari et al. [2022]

---

> ### Author Rebuttal · Authors · 2025-07-30
>
> We thank the reviewer for their constructive feedback. We are glad that you found our results interesting. Here is our response to your questions:
>
> Weakness 1:
>
> Thank you for the suggestion. We would like to mention that the guarantees for Esfandiari et al. and Komiyama et al. are all in the stochastic bandits setting, and their algorithms have $\omega(T )$ regret bounds in the adversarial setting.
>
> Question 1:
>
> That is a great point. In fact, Kalai and Vempala also show that if the randomness $p$ in Algorithm 1 is re-drawn in each round, we can achieve sublinear regret even in the case of adaptive adversaries. Since our algorithms are a generalization of FLL by Kalai and Vempala that observation holds in our case as well. We add a remark to highlight that.
>
> Question 2:
>
> What we mean here is that if we consider the FTPL (and not FTPL with block updates, i.e. FTPLB), over two different cost sequences $S_1$ and $S_2$ that are both drawn from the same sequence $D_1,\cdots, D_T$, then FTPL changes its actions in near timesteps when executing it on $S_1$ and $S_2$. Hence, since with high probability these changes are in close-by timesteps, they will be in the same blocks (whp). Now, if we consider FTPLB, since it is updating its action only at the endpoints of the blocks, it will pick the same action sequence when executing it over $S_1$ and $S_2$.
>
> Question 3:
> You are correct, we will add this to the model section. In Thm D.3, we explain that the $\ell_1$ length of c is at most and the $\ell_1$ diameter of A is D.
>
> Question 4:
>
> That’s a great point; we will enhance the writing of this example. We just mean that initially, when $c_1$ is drawn, that is a vector of length 2 (since we have two actions), the cost for action $a_1$ is higher than the cost of $a_2$ (this is what we mean by bonus, a lower initial cost). However, in hindsight, $a_1$ is a better action, so it would take some time for the algorithm to figure this out.
>
> Last comment:
> Perfect, thanks, we will correct this.

---

> > ### Comment · Reviewer_Pyha · 2025-08-03
> >
> > Thanks for your feedback, the points make sense to me. I find the results of the paper quite interesting and will retain my score.

---

### Official Review · Reviewer_Cdde · 2025-07-03

**Clarity:** 3
**Significance:** 3
**Originality:** 3
**Rating:** 5
**Confidence:** 3

**Summary:**

The paper studies replicable learning for online learning optimization and the experts setting under the online setting. Previous works studied a similar problem in the online setting with data being generated in an iid fashion from a single distribution whereas this work generalizes the setting to allow data points at different timesteps to be sampled from different distributions (i.e. adversarially chosen). The primary objective of this work is to produce algorithms that fit certain replicability properties along with achieving low regret in the online setting.

The authors explain that an online learning algorithm is adversarially $\rho$-replicable to indicate that with probability $1 - \rho$, and for two sequences $S$ and $S'$ generated iid from an adversarially chosen sequence of distributions, the algorithm generates the same set of actions at all timesteps for both sequences (fixing the algorithm's internal randomness).

The paper introduces many results for online replicable learning with the property of adversarially $\rho$-replicability being satisfied. In both online linear optimization and the experts problem, the authors construct two algorithms, one for each setting, that satisfy adversarial replicability along with achieving sublinear regret. More specifically, in online linear optimization, the authors develop an algorithm called the FLLB which is the FLL algorithm executed on the time horizon divided into blocks where the updates happen at the end of each block. Similarly, in the experts problem, the authors take the FTPL and apply the same blocking procedure and use geometric noise instead of the standard exponential noise used by Kalai and Vempala in their FTPL algorithm. They also study regret lower bounds and show a lower bound of $\Omega(\frac{\sqrt{T}}{\rho})$ and extend their results for iid replicability as well. They additionally introduce a general framework to convert any online regret minimization algorithm to one that has a slightly worse regret guarantee but with adversarially $\rho$-replicability satisfied.

**Questions:**

- Regarding the lower bounds for adversarially replicable algorithms, could it be the case that the cost of replicability must exceed the classical $\sqrt{T}$ barrier as seen in traditional online learning? Or is it possible that one could further refine the upper bounds to make them sharper?
- While the authors show that one can preemptively select the block length $B$ beforehand to get the optimal guarantees, is there an adaptive, data-dependent way to choose the block length that would still preserve replicability?

**Ethical Concerns:**

["NO or VERY MINOR ethics concerns only"]

**Final Justification:**

The authors have addressed all my questions during the rebuttal period. As pointed in other reviews, certain modifications need to be made to the paper to make it more technically rigorous. For example, one reviewer found an important condition that needs to be mentioned in Theorem 5.1. Assuming the authors make these modifications to their paper, I still my retain my original rating of a 5 for this paper.

**Limitations:**

Yes

**Quality:**

3

**Strengths And Weaknesses:**

Strengths:
- The paper tackles a unique setting of a more generalized version of replicability under the online setting, namely adversarial $\rho$-replicability.
- The paper provides a pretty comprehensive treatment of this setting with many different results across online linear optimization, the experts problem, an online-to-replicable algorithm reduction, and derivation of lower bounds under each scenario.
 - They also show a nearly optimal iid replicable algorithm in the expert setting with a bound that matches the lower bound of Impagliazzo et al. (2022) up to logarithmic factors.
- The paper is also quite well-written. The authors introduce the setting in an effective manner using tangible examples, give easy-to-follow overviews of their algorithmic techniques, and do a good job of explaining technical details.

Weaknesses:
- Not necessarily a major weakness but there exists a rather noticeable gap between the upper and lower regret bounds, $T^{5/6}$ vs $\sqrt{T}$, under adversarially replicable algorithms. A small discussion, if available, discussing the challenges in closing the gap or why the gap exists would be interesting.

---

> ### Author Rebuttal · Authors · 2025-07-30
>
> We thank the reviewer for their constructive feedback. We are glad that your found our results interesting. Here is our response to your questions:
>
> Question 1:
>
> Regarding the upper bound, our current algorithm would not achieve a better bound. The reason is that, there are two variables in our algorithms, one the block size B, and the other one, the noise level $\epsilon$. In order to bound the probability of non-replicability to at most $\rho$, we need to set these two parameters carefully (equation below line 853). Furthermore, in order to minimize regret, we need to set $\epsilon = \frac{1}{\sqrt{BT}}$, line 856. Putting these not inequalities together, gives us the optimum values of noise level $\epsilon$ and block size B. However, there might be a different algorithm that achieves a better regret bound, and we leave it as an open question.
>
> On the other hand, regarding the lower bound, currently, all approaches to proving lower bounds in terms of the replicability parameter go via a reduction to the coin problem as defined in [Impagliazzo et al. STOC’ 2022](Lemma 7.2). (Reduction for the stochastic bandits mentioned somewhat informally in Esfandiari et al.) The coin problem is as follows: promised that a 0−1 coin has bias either $1/2−\tau$ or $1/2+\tau$ for
> fixed $\tau > 0$ how many flips $T$ are required to identify the coin’s bias with high probability while being $\rho$-replicable? Here we have a tight lower bound of $\rho.\tau.\sqrt{T} > \Omega(1)$. When the reduction is done (as in the paper) the $\sqrt{T}$ here translates to the $\sqrt{T}$ in our proof. We don’t believe that this reduction is optimal in the sense that a more batched up division of the time steps where each batch reduces to a coin problem with different $\tau$ values should lead to a better bound: however, we are unable to execute the above idea. This would also mirror the batching of time steps in the upper bound, and would utilize the full extent of the adversarial setting where the distributions (which correspond to $\tau$ in the reduction) are time varying.
>
> Question 2:
>
>  That is a great question. We derived these results so that they would capture both adversarial and iid input sequences, as well as their mixtures, which can be modeled by setting certain distributions as point-masses. However, in the iid case, in Appendix H, we show how to derive better regret bounds $O(\sqrt{T})$ by having blocks of varying lengths.
> However, it is certainly possible that our $T^{5/6}$ regret bound in the general case is improvable, and we leave it as an open question.

---

> > ### Comment · Reviewer_Cdde · 2025-08-04
> >
> > I thank the authors for taking the time to respond to my review.
> >
> > Thank you for clarifying my question regarding the lower bound. As a quick comment, in Section 7, the reduction to the coin problem described in Section 7 simply states the coin problem from Impagliazzo et al. and doesn't include the requirement for being $\rho$-replicable. Is this being implied implicitly? Or does effectively answering the coin problem give you $\rho$-replicable algorithm?
> >
> > Overall, the authors have addressed all my main questions, and I believe that this paper should be accepted. Therefore, I retain my original score of a 5.

---

> > > ### Author Response · Authors · 2025-08-04
> > >
> > > Thank you for your response. We understand the point of confusion, here we were just explaining the coin problem. The exact lemma from Impagliazzo et al is as follows and is mentioned in Lemma I.1.: "Let $ \tau < \frac{1}{4}$ and  $ \rho < \frac{1}{16}$. Let $ \mathcal{B} $ be a $\rho$-replicable algorithm that decides the coin problem with success probability at least $ 1 - \delta$ for  $\delta = \frac{1}{16}$ using $T$ iid samples.
> > >  Furthermore, assume $\mathcal{B}$ is $\rho$-replicable, even if its samples are drawn from a coin $\mathcal{C}$ with bias in $\left( \frac{1}{2} - \tau, \frac{1}{2} + \tau \right)$. Then $\mathcal{B}$ requires sample complexity $T \in \Omega\left( \frac{1}{\tau^2 \rho^2} \right)$, i.e., $\rho \in \Omega\left( \frac{1}{\tau \sqrt{T}} \right)$."
> > >
> > > We will clarify this in the text.

---

### Decision · Program_Chairs · 2025-09-17

**Decision:**

Accept (poster)

**Comment:**

This paper considers online learning in setting between stochastic and adversarial: they consider adversarial, but oblivious, sequences of distributions as a comparator class under a rho-replicability assumption. They develop non-trivial regret bounds in a variety of settings. Reviewers found hthe treatment of the setting fairly complete, completing with a matching (up to log terms) bound in the i.i.d. setting.
Reviewers also praised the presentation and clarity of the writing.

There was some concern about the proof of the main theorem, but this seems to have be dealt with during the discussion period. A few points about readability were also discussed, and I hope the authors use the discussion results to improve the quality of the paper.

Given the complehensive results, interesting setting, and good presentation, this was an easy paper to accept.